# Break-induced replication promotes formation of lethal joint molecules dissolved by Srs2

Rajula Elango[1], Ziwei Sheng[2], Jessica Jackson[3], Jenna DeCata[1], Younis Ibrahim[1], Nhung T. Pham[4], Diana H. Liang[4,5], Cynthia J. Sakofsky[1,6], Alessandro Vindigni[3], Kirill S. Lobachev[2], Grzegorz Ira[4] & Anna Malkova [1]

Break-induced replication (BIR) is a DNA double-strand break repair pathway that leads to genomic instabilities similar to those observed in cancer. BIR proceeds by a migrating bubble where asynchrony between leading and lagging strand synthesis leads to accumulation of long single-stranded DNA (ssDNA). It remains unknown how this ssDNA is prevented from unscheduled pairing with the template, which can lead to genomic instability. Here, we propose that uncontrolled Rad51 binding to this ssDNA promotes formation of toxic joint molecules that are counteracted by Srs2. First, Srs2 dislodges Rad51 from ssDNA preventing promiscuous strand invasions. Second, it dismantles toxic intermediates that have already formed. Rare survivors in the absence of Srs2 rely on structure-specific endonucleases, Mus81 and Yen1, that resolve toxic joint-molecules. Overall, we uncover a new feature of BIR and propose that tight control of ssDNA accumulated during this process is essential to prevent its channeling into toxic structures threatening cell viability.

[1] Department of Biology, University of Iowa, Iowa City, IA 52242, USA. [2] School of Biological Sciences and Institute for Bioengineering and Bioscience, Georgia Institute of Technology, Atlanta, GA 30332, USA. [3] Edward A Doisy Department of Biochemistry and Molecular Biology, Saint Louis University School of Medicine, St. Louis, MO 63103, USA. [4] Department of Molecular & Human Genetics, Baylor College of Medicine, Houston, TX 77030, USA. [5] Present address: Houston Methodist Hospital, 6565 Fannin St., Houston, TX 77030, USA. [6] Present address: Genome Integrity and Structural Biology Laboratory, National Institute of Environmental Health Sciences, US National Institutes of Health, Research Triangle Park, Durham, NC 27709, USA. Rajula Elango and Ziwei Sheng contributed equally to this work. Correspondence and requests for materials should be addressed to K.S.L. (email: kirill.lobachev@biology.gatech.edu) or to G.I. (email: gira@bcm.edu) or to A.M. (email: anna-malkova@uiowa.edu)

DNA double-strand breaks (DSBs) is a lethal DNA damage if it remains unrepaired. Break-induced replication (BIR) is one pathway to repair DSBs where only one broken end can invade into the homologous template, which is similar to breaks formed by collapsed replication forks or eroded telomeres. BIR promotes massive genomic instability including hyper-mutagenesis, formation of mutation clusters and gross chromosomal rearrangements similar to those leading to cancer[1–6]. Recent data obtained in human cells showed that BIR can be induced by oncogene overexpression[7] or by replication stalling at chromosome fragile sites[8] and can potentially serve as a trigger for carcinogenesis. In addition, 10–15% of cancer cells maintain their telomeres through alternative lengthening of telomere (ALT), a BIR-like pathway[9]. This makes it essential to understand the mechanism of BIR and to specifically identify the steps of BIR that can serve as potential targets for anti-BIR therapy.

Similar to other homologous recombination pathways, BIR initiates by 5′-3′ resection of a DSB end which then invades into the homologous template and initiates synthesis that can copy > 100 kb of the template till the end of the chromosome. Despite extensive DNA synthesis, BIR mechanism is very distinct from S-phase DNA replication. Instead of a replication fork, BIR is carried out by a migrating bubble, which leads to conservative inheritance of newly synthesized DNA[10–12]. Another important difference between S-phase replication and BIR is that during BIR, leading and laggings strands are synthesized in an asynchronous manner, and this leads to accumulation of long ssDNA regions[10]. These regions are stabilized by RPA, ssDNA binding protein[13]. Another protein that binds long ssDNA accumulated in the course of BIR is Rad51, a strand exchange protein. However, it remains unknown how cells control this binding, as well as the dynamics of the resulting Rad51 nucleofilament to avoid its unscheduled pairing with a template (e.g., homologous chromosome) that can threaten genomic stability. To address this question, we decided to test the role of known Rad51 regulators in BIR.

A known regulator of Rad51 nucleofilament is Srs2 helicase. Both genetic and biochemical evidence demonstrated that Srs2 can remove Rad51 from ssDNA, an activity that limits spontaneous recombination[14–19]. In addition, a number of other genetic studies proposed an anti-recombination role played by Srs2[14–17, 20–31] and also by its bacterial functional homolog, UvrD[32]. Two domains of Srs2 are required for stripping of Rad51 from ssDNA (so called "strippase" activity): the translocase and Rad51-interacting domains. The translocase (motor) domain of Srs2 resides in N- terminal portion of the protein and can be disrupted by a K41A mutation, inactivating ATPase activity of the protein[33]. The Rad51-interaction (BRC) domain resides in the C-terminal part of Srs2 and is disrupted in srs2-BRCΔ lacking 862–914 residues[34]. Another biochemical function of Srs2 is in its helicase activity. This activity can unwind various DNA substrates[35–38], but the role of the helicase activity for the anti-recombination function of Srs2 remains unclear.

We note that previous work suggested a pro-recombinogenic role for Srs2 in DSB repair pathways via homologous recombination (HR). Srs2 was reported to promote DSB repair by synthesis-dependent strand annealing (SDSA) and to limit crossover outcomes[39, 40]. Also, Srs2 was proposed to be involved in DNA damage checkpoint recovery[41, 42], and this function was deduced from the observation of massive cell death following completion of DSB repair. In the most extreme case, 98% of srs2Δ cells died despite of what appeared as successful completion of DSB repair by single-strand annealing (SSA), as detected by Southern blot analysis[41]. Later, this death was attributed to the incomplete unloading of recombination factors leading to persistent binding of Rad51 and RPA to the ssDNA surrounding the areas of repair even after the repair has been completed[43, 44]. Together these studies proposed different pro-recombination roles of Srs2 in DSB repair.

In this study, we provide the evidence for a different "anti-toxic" role Srs2 plays to ensure successful DSB repair. Here, we report that bubble migration during BIR requires Srs2. We demonstrate that in the absence of Srs2, most cells fail to complete BIR and die. This massive death results from accumulation of unresolved toxic joint molecules that are formed by invasion of long ssDNA into the intact donor, which leads to trapping of donor and recipient chromosomes together and interferes with BIR completion. We propose that two main activities of Srs2, strippase and helicase, counteract toxic joint intermediates by preventing their formation and promoting their disruption, respectively. Overall, our findings demonstrate that ssDNA produced by bubble-migration represents a vulnerable intermediate of BIR that could be lethal for the cell, and we speculate that this property can be used in the future to specifically target cells undergoing BIR.

## Results

**Srs2 counteracts toxic joint molecules formed during BIR**. To examine whether BIR promotes the formation of toxic recombination intermediates in the absence of Srs2, we used a budding yeast experimental system where a DSB is repaired by BIR involving homologous chromosomes[45] (Fig. 1a, Supplementary Table 1). In this system, a galactose-inducible DSB is initiated by HO endonuclease at the MATa locus of the truncated copy of chromosome III (Chr. III), while the full-length donor copy of Chr. III contains an uncleavable MATα-inc allele and serves as the template for DSB repair. Elimination of all but 46 bp of homology on one side of the break on the recipient molecule via replacement with LEU2 and telomere sequences results in efficient DSB repair through BIR. Initiation of BIR in this system is preceded by extensive 5′-to-3′ resection of the DSB at MATa[46], and is followed by strand invasion of the 3′ single-strand end into the donor chromosome at a position proximal to the Yα-inc sequences and the subsequent copying of ~100 kb of donor DNA to the right telomere (Fig. 1a). BIR and alternative repair outcomes can be followed based on maintenance of markers located at all arms of recombining molecules (Fig. 1b).

Here, using this BIR assay, we observed that srs2Δ mutant cells exhibited a large loss of viability (Fig. 1c), while BIR remained the predominant repair outcome among srs2Δ survivors (Fig. 1d). BIR events obtained in srs2Δ, however, differed from those in wild- type (SRS2) cells with respect to mutagenesis that is normally high in association with BIR[47]. The level of frameshifts measured using lys2-$A_4$ reporters placed on the track of BIR was reduced five-fold and four-fold for 16 kb and 36 kb positions of the reporter, respectively, in srs2Δ as compared to wild type (SRS2) cells (Supplementary Fig. 1a). The level of base substitutions, measured using ura3-29 reporter (similar to ref. [10]) was also reduced five-fold and seven-fold for two different orientations of the reporter (Supplementary Fig. 1b). This reduced mutagenesis suggested that the absence of Srs2 affected BIR progression among the survivors as well.

Massive death observed in srs2Δ mutants was atypical for our experimental system, where the presence of two copies of chromosome III, one of which remains unbroken, allows the cells to survive even in situations when a second, broken chromosome is unrepaired and lost (Fig. 1b), e.g., in rad52Δ mutants[48]. Thus, the loss of viability suggested that initiation of BIR in the absence of Srs2 compromises not only the broken chromosome but also the intact donor serving as a template for

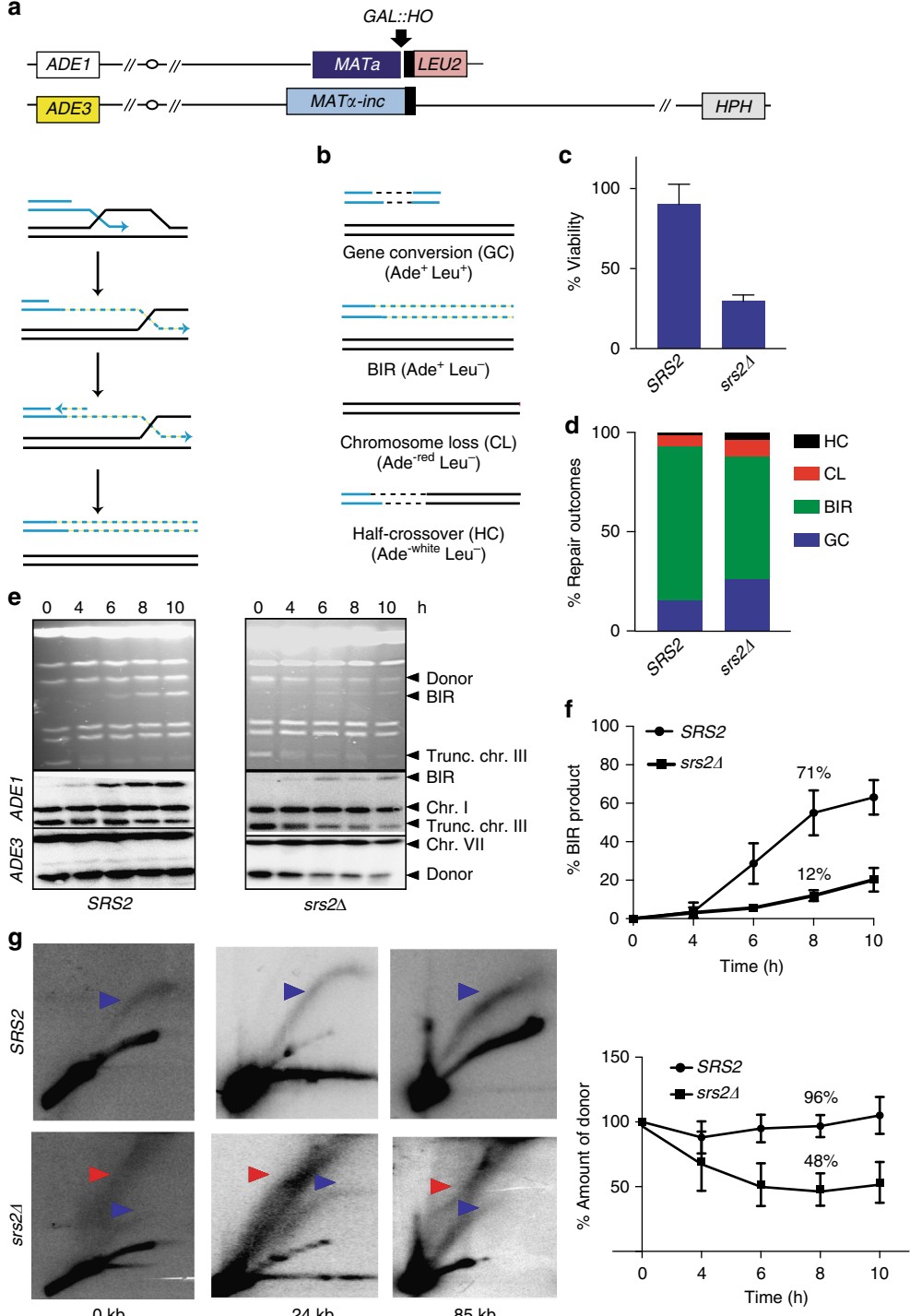

**Fig. 1** BIR leads to the formation of toxic joint molecules counteracted by Srs2. **a** BIR is initiated by DSB introduced by HO endonuclease at *MAT*a locus in yeast disomic for Chr. III. The broken recipient chromosome (blue) invades unbroken homologous donor (black). Repair DNA synthesis is initiated and progresses by a migrating bubble, which leads to the conservative inheritance of newly synthesized DNA. **b** DSB repair outcomes with corresponding phenotypes (in parenthesis). **c** Cell viability following DSB induction (%). **d** Distribution of DSB repair outcomes in *SRS2* (WT) and *srs2*Δ based on their phenotypes (refer to (b)). **e** BIR kinetics analyzed by CHEF gel using cells taken at indicated time points following DSB induction. Upper panel: CHEF gels stained with Ethidium Bromide. Subsequent panels below show Southern blot analysis using *ADE1*-specific, and *ADE3*-specific probes as indicated. **f** Quantification of BIR product (top) and of donor chromosome entering the gel (bottom). **g** 2D gel analysis of BIR intermediates in *SRS2* (top panel) and *srs2*Δ (bottom panel) at 7 h following DSB induction. Genomic DNA was digested with KpnI to detect intermediates at 0 kb and with BglII to detect intermediates at 24 kb and 85 kb. Intermediates were detected using probes specific to the following positions on Chr. III: 0 kb (left panel), 24 kb (middle panel) and 85 kb (right panel) away from the DSB. Blue arrowheads denote bubble arc intermediate and red arrowheads denote 'rubble' structure. Quantification of cell viability and DSB repair efficiency was based on results of ≥ 3 independent experiments and are presented as means ± s.d

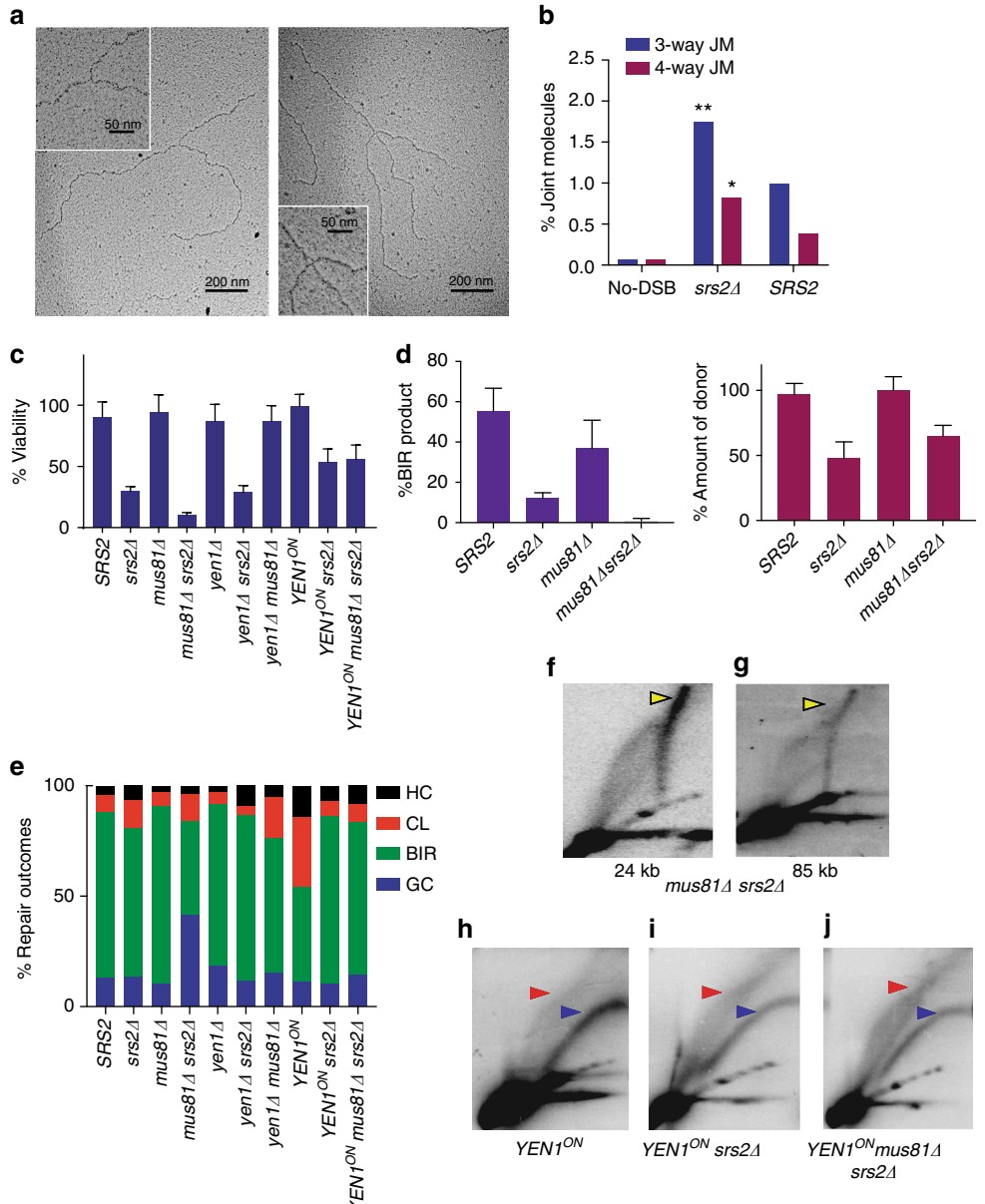

**Fig. 2** Structure-specific endonucleases resolve toxic BIR intermediates. **a** Representative images of DNA fragments containing 3-way and 4-way junctions (joint molecules, JMs) in *srs2Δ* identified by EM analysis of genomic DNA isolated 7 h following induction of BIR. Scale bars correspond to 200 nm in the entire field images and 50 nm in the enlarged images. **b** The fraction of DNA fragments containing 3-way and 4-way junctions in *SRS2*, *srs2Δ* and no-DSB *srs2Δ* control. The ** and * indicate a statistically significant difference from *SRS2* with *P* = 0.0066 and *P* = 0.0379, respectively. **c** Cell viability following BIR induction in cells containing various mutations, including *srs2Δ*, *mus81Δ*, *yen1Δ*, and *YEN1ON*. **d** Quantification of BIR product (left) and of the donor chromosome (right) at 8 h following initiation of BIR. **e** Distribution of DSB repair outcomes following BIR repair in various mutants shown in C. **f** and **g** 2D gel analysis of BIR intermediates in *srs2Δ mus81Δ* using probes specific to 24 kb position and 85 kb position, respectively, away from the DSB site. **h–j** shows 2D intermediates detected by 24 kb probe in **h** *YEN1ON*, **i** *YEN1ON srs2Δ*, **j** *YEN1ON mus81Δ srs2Δ*. Yellow and blue arrowheads denote "spike" and bubble intermediates, respectively. Red arrowhead denotes 'rubble' intermediate. Quantification of cell viability, the amount of product formed, and amount of donor present was based on ≥ 3 independent experiments and presented as means with standard deviations (s.d.)

repair. To test this idea we analyzed BIR progression using chromosome-separating CHEF gel electrophoresis and observed two striking defects in *srs2Δ*. First, the amount of BIR product measured 8 hours (h) after DSB induction was nearly five-fold less than in wild-type (Fig. 1e, f). Second, while the amount of template (donor) Chr. III molecule in wild-type cells remained constant throughout the course of BIR, in *srs2Δ* it drastically decreased. At 8 h, the amount of the donor entering the gel was only 48% of the initial amount before DSB induction in *srs2Δ* as compared to 96% in *SRS2* (Fig. 1e, f).

We hypothesized that the decrease of donor molecules in the agarose gel in *srs2Δ* results from accumulation of recombination intermediates as branched DNA structures. This was tested using 2D gel electrophoresis of restriction enzyme digested genomic DNA obtained from *SRS2* and *srs2Δ* cells undergoing BIR. We have previously used this method to demonstrate that BIR is carried out by a migrating bubble, and ssDNA accumulates due to asynchronous synthesis of leading and lagging strands[10]. Following 2D gel electrophoresis, DNA digested with BglII or KpnI (see legend to Fig. 1g) was hybridized with radioactively labeled

probes specific to various positions located 0 kb, 24 kb and 85 kb centromere-distal to the DSB position (Fig. 1g). BIR in wild-type cells was associated with the formation of bubble-arc replication

intermediate that was previously described[10]. In srs2Δ, the bubble intermediate (blue arrow) was barely detectable while another BIR intermediate became more prominent. This intermediate

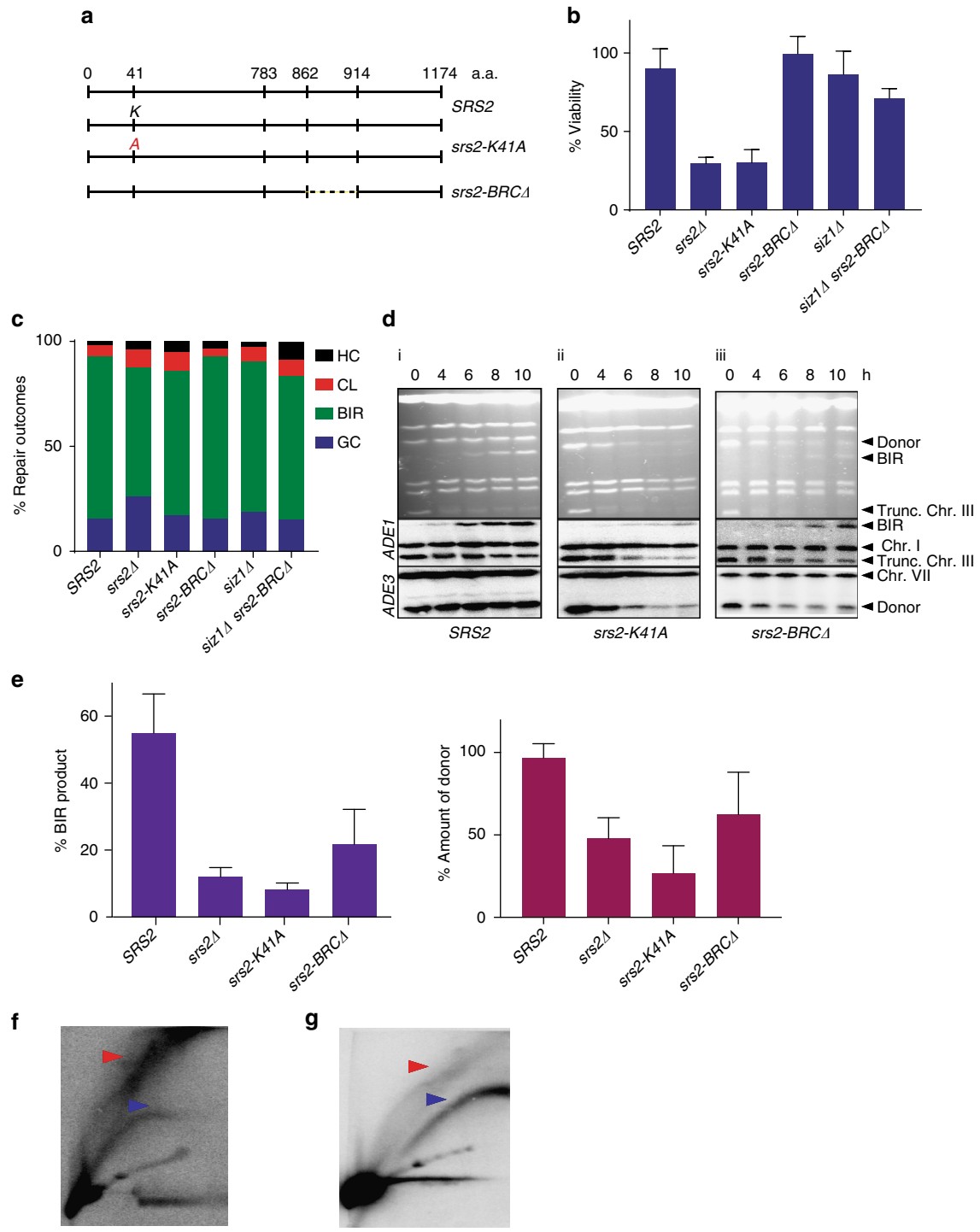

**Fig. 3** The defects in the helicase and strippase activities of Srs2 interfere with BIR progression. **a** The schematic of Srs2 (WT) and of *srs2* mutations *srs2-K41A* and *srs2-BRCΔ*. "A" represents the position where lysine is replaced by alanine resulting in the ATPase (strippase and helicase) defect. Dotted line denotes deletion of *SRS2 BRC* domain. **b** Cell viability following DSB induction in *SRS2*, *srs2Δ*, *srs2-K41A*, *srs2-BRCΔ*, *siz1Δ* and *siz1Δ srs2-BRCΔ*. **c** Distribution of DSB repair outcomes following BIR repair in various mutants shown in **b**. **d** BIR kinetics in *srs2-K41A* and *srs2-BRCΔ* analyzed by CHEF (see Fig. 1e for details). **e** Quantification of BIR product (left) and of donor entering the gel (right) measured at 8 h after DSB. **f**, **g** 2D analysis of BIR in *srs2-K41A* and *srs2-BRCΔ*, respectively A probe specific to 24 kb away from the DSB was used. Blue and red arrowheads denote bubble arc intermediate and 'rubble' structure respectively. Quantification of cell viability, amount of BIR product, and amount of donor was based on ≥ 3 independent experiments and are presented as means with standard deviations (s.d.)

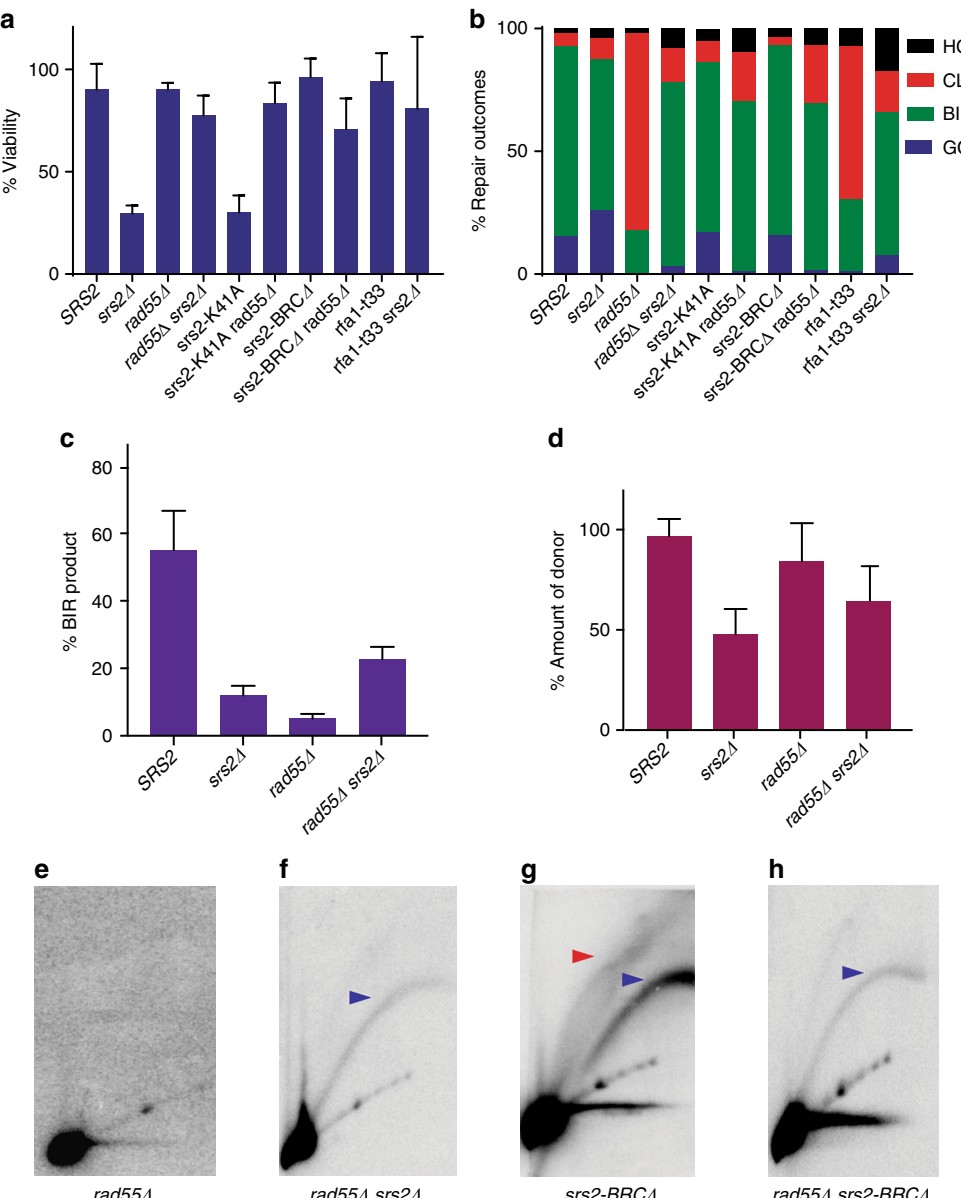

**Fig. 4** Destabilization of Rad51 filament bypasses the requirement for Srs2 during BIR. **a** Cell viability following DSB induction in *rad55Δ*, *rfa1-t33* and *srs2Δ* mutants. **b** distribution of repair outcomes following DSB induction in *rad55Δ*, *rfa1-t33* and *srs2Δ* mutants. Quantification of BIR product **c** and of the donor entering the gel **d** measured at 8 h following DSB repair initiation. **e-h** 2D gel analysis of BIR intermediates in the following *rad55* and *srs2* mutants: **e** *rad55Δ*, **f** *rad55Δ srs2Δ*, **g** *srs2-BRCΔ*, and **h** *rad55Δ srs2-BRCΔ*. Blue and red arrowheads denote bubble arc intermediate and "rubble" structure respectively. Quantification of cell viability, amount of BIR product, and amount of donor was based on ≥ 3 independent experiments and are presented as means with standard deviations (s.d.)

(Fig. 1g; red arrow) consisted of heterogeneous DNA molecules that were more branched and heavier than those forming the bubble intermediate and we will refer to this intermediate as 'the rubble'.

We propose that long stretches of ssDNA generated by DSB resection and DNA synthesis can promote formation of additional joint structures located behind the migrating BIR bubble and interfere with BIR completion. This hypothesis of joint molecule formation was tested by electron microscopy (EM) analysis of whole genome from DNA extracted from *srs2Δ* and *SRS2* cells, digested with BglII and enriched for ssDNA-containing fragments by passing through a BND cellulose column (Fig. 2a, b; Supplementary Fig. 2 and 3). We observed a significant increase (*P* = 0.0003) of branched DNA molecules,

which included 4-way and 3-way junctions in *srs2Δ* (100/3860 molecules) as compared to *SRS2* cells (47/3395 molecules) (Supplementary Table 2). This was consistent with accumulation of joint molecules (JMs) in *srs2Δ*. As expected, the level of branched molecules was very low in a no-DSB control (Fig. 2b).

**BIR toxic JMs are resolved by structure-specific nucleases.** Based on the large amount of unusual joint molecules observed during BIR in *srs2Δ* cells, we hypothesized that DSB repair survivors in *srs2Δ* cells may require nucleases that cleave JMs. Towards this goal we tested the effect of elimination of Mus81-Mms4 from *srs2Δ* cells and activation of Yen1 that is normally inactive in G2. Both Mus81 and Yen1 are known to process

Holliday junctions (HJ), and other branched intermediates (reviewed in ref. [49]). Elimination of Mus81 in *srs2Δ* led to further decrease of cell viability upon DSB induction: from 30% in *srs2Δ* to only 10% in *srs2Δ mus81Δ* (Fig. 2c). Deletion of *MUS81* itself does not affect repair or viability following BIR induction[12] (Fig. 2c and Supplementary Fig. 4). In addition, the amount of BIR product accumulated in *srs2Δ mus81Δ* at 8 h following the break (as detected by CHEF) was 10-fold lower as compared to *srs2Δ* (Fig. 2d). Based on these results, we propose that Mus81 resolves some of the toxic JMs accumulated in the absence of Srs2. In support of this conclusion, we found that the structures observed by 2D gel electrophoresis of BIR intermediates accumulated in *srs2Δ mus81Δ* looked different from the 'rubble' structures. In particular, the intermediates accumulated in *srs2Δ mus81Δ*, contained molecules of larger mass than those in the "rubble" structure (Fig. 2f, g; yellow arrows) and were reminiscent of the 'spike' structures that were previously reported for joint molecules in meiosis[50–52]. We propose that these structures represent a high molecular weight and highly branched intermediate corresponding to the multi-invasion regions formed behind the bubble (See Supplementary Fig. 5 and discussion for details). We envision that partial resolution of these multi-invasion intermediates by Mus81 leads to formation of transient intermediates (Supplementary Fig. 6) that are visualized as rubbles on 2D gel electrophoresis. These transient intermediates can be processed further which allows formation of viable BIR outcomes.

The role of another structure-specific endonuclease Yen1 was not evident based on the viability data of *yen1Δ srs2Δ* double mutant (Fig. 2c). This is expected because Yen1 nuclease is activated only in late M-phase[53, 54]. In order to test the ability of Yen1 to resolve toxic DNA structures formed in *srs2Δ*, we used a constitutively active allele *YEN1*[ON][53]. We observed that expression of *YEN1*[ON] largely suppressed the defect of both *srs2Δ* and *srs2Δ mus81Δ* mutant cells, which suggested that Yen1[ON] is capable of processing toxic intermediates formed in these mutants and the resolution of these intermediates helps survival (Fig. 2c). Consistently, *YEN1*[ON] *srs2Δ mus81Δ* accumulated a "rubble" intermediate (similar to the one observed in *srs2Δ*) instead of a 'spike' structure that was observed in *srs2Δ mus81Δ* (Fig. 2h–j). Interestingly, expression of *YEN1*[ON] in wild-type cells resulted in decreased BIR and an increase in chromosome loss (CL) and half-crossover (HC) events (Fig. 2e; 1b). However, the formation of CL and HC events were very rare following BIR initiation in *YEN1*[ON] *srs2Δ* strains (Fig. 2e). We note that the abnormal repair observed in *SRS2 YEN1*[ON] cells is in agreement with the view that Yen1 should be normally inactive at G2 when BIR occurs. In addition, using the same strains, we observed that *YEN1*[ON] partially suppressed MMS sensitivity of *srs2Δ* (Supplementary Fig. 7a), which suggested that the molecular function of Srs2 that we discovered for BIR might be involved in repair of MMS-induced DNA damage as well.

**Two detox functions of Srs2 during BIR**. We propose that Srs2 promotes BIR by counteracting toxic intermediates via its two anti-recombinational activities: a "strippase" activity, which removes Rad51 from ssDNA thus preventing joint molecule formation and also through its helicase activity (reviewed in ref. [55]) which physically disrupts the toxic joint intermediates. To test this idea we examined the effects of mutations in two Srs2 domains: (i) the ATPase domain (*srs2-K41A*) required for stripping Rad51 and DNA unwinding[33]; and (ii) the BRC-domain (*srs2-BRCΔ*) required for Rad51 interaction and for efficient stripping of Rad51 from ssDNA[34] (Fig. 3a). The effect of *srs2-K41A* mutation on BIR was similar to that of *srs2Δ* with respect to

cell survival (Fig. 3b), and the amount of BIR product formation (Fig. 3d, e). In addition, similar to *srs2Δ*, a large fraction of the *srs2-K41A* donor molecules failed to enter the CHEF gel following DSB induction (Fig. 3d, e), and accumulated 'rubble' intermediates as monitored by 2D gel electrophoresis (Fig. 3f). Thus, the ATPase activity of Srs2, which is required for both its strippase and helicase activities, is necessary for BIR.

Next, we tested the importance of the BRC domain of Srs2 (encompassing residues 862–914 (Fig. 3a)) in BIR. We expected that the absence of this region would affect the ability of Srs2 to remove Rad51 from the filament, but not its helicase activity[34, 35]. We proved this for our strain background by confirming that *srs2-BRCΔ* suppressed the MMS sensitivity of *rad18Δ* (Supplementary Fig. 7b) similar to other *srs2* mutants that were deficient in the "strippase" activity but possessed helicase activity[34]. The BIR assay performed in *srs2-BRCΔ* demonstrated successful BIR completion with the viability of ~ 90% (Figs. 3b and 4a), and with BIR outcomes comprising 80% of the survivors (Fig. 4b). At the same time, the physical analysis of BIR using CHEF indicated slower kinetics of BIR with only 22% of BIR product accumulated in *srs2-BRCΔ* at 8 h vs. 55% in *SRS2* cells (Fig. 3d, e). Also, a significant fraction of the donor failed to enter the agarose gel during CHEF analysis in the middle of BIR time course (Fig. 3d, e). These defects were indicative of branched structure accumulation. Indeed, 2D gel electrophoresis detected "rubble" intermediates in *srs2-BRCΔ* cells (Fig. 3g). Based on these results we propose that the defect of *srs2-BRCΔ* in removal of the Rad51 from the filament[34] leads to the formation of toxic intermediates that are eventually removed by the helicase activity of Srs2, allowing slower but efficient repair and survival.

**Unstable Rad51 filament bypasses the need for Srs2**. We hypothesized that the need for Srs2 could be bypassed by having less stable Rad51 nucleofilament. Towards this goal, we used the *rad55Δ* mutant where Rad51 filament is inherently unstable[30]. Deletion of *RAD55* in *srs2Δ* led to complete recovery of BIR and cell viability (Fig. 4a, b). The viability of *srs2Δ rad55Δ* cells was ~ 80%, which is much higher than in *srs2Δ* (30%), and 75% of the survivors in *srs2Δ rad55Δ* resulted from BIR, which was significantly higher compared to *rad55Δ* single mutant (17%). The latter observation suggests that Srs2-dependent removal of Rad51 in the absence of Rad55 prevents successful strand invasion, which leads to chromosome loss.

Similarly, combining *rad55Δ* and the helicase-dead mutant, *srs2-K41A*, led to high cell viability (to ~ 80%) and efficient (~ 70%) BIR (Fig. 4a, b). This result was further confirmed by 2D gel electrophoresis, where normal BIR bubble-arc, and no "rubble" intermediates were observed in *rad55Δ srs2Δ* and *rad55Δ*, respectively (Fig. 4f, e). These results suggest that when the formation of toxic intermediates is precluded by instability of Rad51 filament, Srs2 is less needed for the completion of BIR than in the case of "excessive" and stable Rad51 filament. While the efficiency of BIR in *srs2Δ rad55Δ* mutants appears to be much better than in single *srs2Δ* or *rad55Δ* mutants, the kinetics of BIR product formation in *srs2Δ rad55Δ* is significantly slower than in wild-type cells, with respectively 23% vs. 55% of product accumulated 8 h following DSB induction (Fig. 4c). This slower BIR kinetics in *srs2Δ rad55Δ* indicates that the maximum efficiency of BIR requires optimized Rad51 filament that can be formed only in the presence of both Srs2 and Rad55. Nevertheless, most *srs2Δ rad55Δ* cells successfully completed BIR. In addition, the *srs2-BRCΔ* allele restored BIR in *rad55Δ* mutants (Fig. 4a, b, h), but the extent of restoration was less as compared to *srs2Δ* (Fig. 4b). We envision that the higher frequency of chromosome losses observed in *srs2-BRCΔ rad55Δ* likely resulted

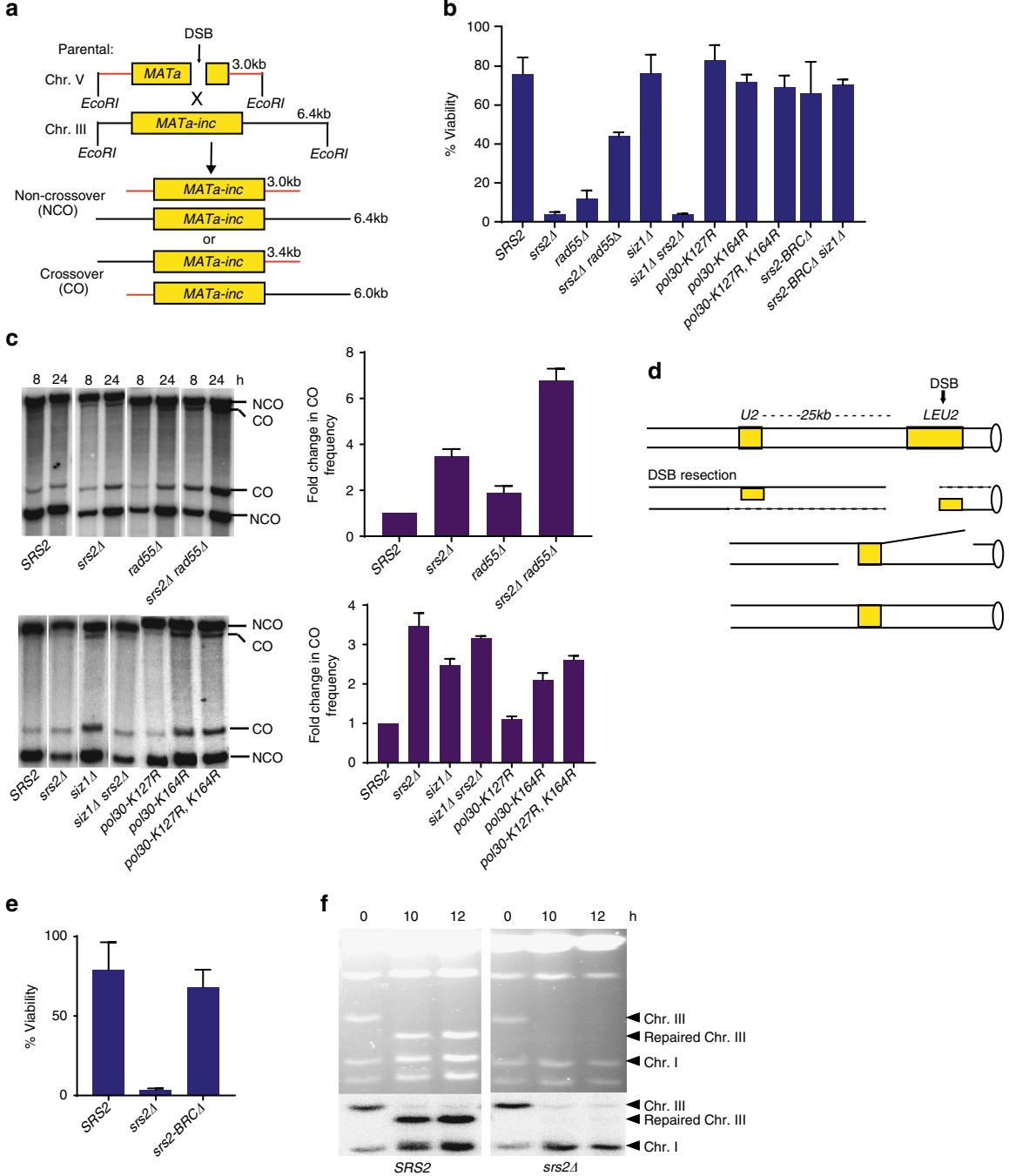

**Fig. 5** Anti-toxic function of Srs2 during various DSB repair pathways. **a** Ectopic gene conversion is initiated by DSB introduced by HO endonuclease at *MATa* inserted at the *ARG5,6* locus on Chr. V and proceeds by using homologous *MATa-inc* sequence on Chr. III as a template. The schematic explains the assay to distinguish crossover from non-crossover products of GC. **b** Cell viability reflecting the efficiency of GC in various mutant backgrounds. **c** Crossover (CO) and non-crossover (NCO) GC products detected by Southern blot analysis (left, top panel) at 8 h and 24 h for *SRS2*, *srs2Δ*, *rad55Δ*, and *rad55Δ srs2Δ*. CO and NCO GC products were also detected at 8 h for various other mutants (left, bottom panel). **d** The schematic of DSB repair system in YMV80 strains where DSB is induced at *LEU2* by HO endonuclease on Chr.III and repaired by SSA/BIR using *U2* as a template. **e** Cell viability in *SRS2* and *srs2* mutants following SSA/BIR. **f** CHEF gels showing analysis of repair by SSA/BIR in *SRS2* (wild type; left) and *srs2Δ* (right). Upper panel: Ethidium Bromide stained gel. Lower panel: Southern blot analysis using *ADE1*-specific probe highlighting Chrs. I and III. Quantification of cell viability and fold-change in CO frequency was based on ≥ 3 independent experiments and are presented as means with standard deviations (s.d.)

from the reversal of the D-loop by the helicase activity of Srs2, which is also called anti-crossover activity (see below).

Recent reports demonstrated that a mutation affecting ssDNA binding protein RPA (*rfa1-t33*) leads to a defect in BIR that may result from decreased Rad51 loading[13]. Thus, we tested whether similar to *rad55Δ*, *rfa1-t33* can suppress low viability of *srs2Δ*. Indeed, viability of *rfa1-t33 srs2Δ* double mutant was much higher when compared to *srs2Δ* (Fig. 4a). Moreover, the

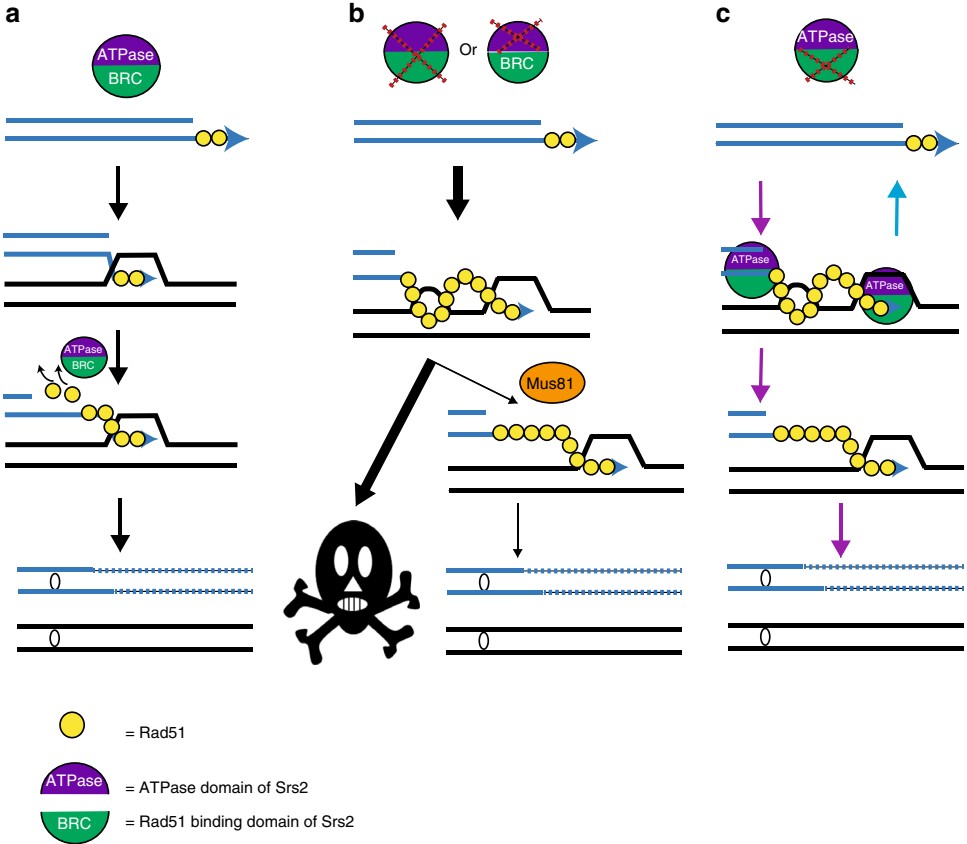

**Fig. 6** The model of BIR associated toxic joint molecules. **a** The strippase activity of Srs2 requiring both the ATPase domain (purple semi-circle) and the Rad51 binding domain (BRC; green) removes excess of Rad51 (yellow circles) from the filament, which prevents formation of toxic intermediates behind the migrating BIR bubble, and ensures efficient BIR. **b** In the absence of Srs2 or of its ATPase activity (*srs2-K41A*) additional Rad51 bound to long ssDNA leads to formation of toxic intermediates, which predominantly lead to cell death (denoted by the skull), but can be resolved by Mus81-Mms4, which allows BIR completion and cell survival. **c** When strippase activity of Srs2 is defective (in the absence of the Rad51 binding domain), toxic intermediates are accumulated, but can be dismantled by helicase activity of Srs2 (pink arrow). In addition, Srs2 can unwind D-loops using its helicase activity (blue arrow)

proportion of cells completing BIR was significantly increased as compared to *rfa1-t33* (Fig. 4b). These results suggested that the defect in the Rad51 filament formation by either *rad55Δ* or *rfa1-t33* suppress the defect of *srs2Δ* in BIR. Overall, our results suggest that the need for Srs2 in BIR can be bypassed by compromising the quality (*rad55Δ*, *rfa1-t33*) of Rad51 filament.

**The detox role of Srs2 in other DSB induced repair pathways.** Having demonstrated the role of Srs2 in counteracting toxic intermediates formed by ssDNA during BIR, we were prompted to ask whether it might play a similar role in other DSB repair pathways including gene conversion and SSA[41].

Gene conversion (GC) was tested in a simple ectopic recombination system where DSB induced within a 1.9 kb *MATa* sequence inserted at the *ARG5,6* locus on Chr. V is repaired by recombination with homologous *MATa-inc* sequence on Chr. III (Fig. 5a)[40, 41]. Repair in this assay proceeds by SDSA with rare ( < 5%) crossover outcomes. It was previously reported that the efficiency of repair, measured 8 h after DSB by Southern blot analyses of the digested genomic DNA, was decreased in the absence of Srs2 from ~ 85% to about ~ 30%. Importantly, cell viability following DSB induction in *srs2Δ* was reduced to just several percents[41] (Fig. 5b). In addition, we observed that *srs2-BRCΔ* did not affect cell viability during GC (Fig. 5b), indicative of the important role of Srs2 motor activity for survival.

As with BIR, the viability of double mutant *srs2Δ rad55Δ* during ectopic GC was significantly improved as compared to *srs2Δ* or *rad55Δ* single mutants (Fig. 5b). This data suggests that destabilization of Rad51 filament by *rad55Δ* improved SDSA repair in *srs2Δ*. This supported the idea that the detox function of Srs2 is required for the successful completion of gene conversion. While the efficiency of GC in *srs2Δ rad55Δ* mutants appears to be much better than in single *srs2Δ* or *rad55Δ*, crossover outcomes were dramatically increased when compared to single mutants (Fig. 5a, c). Thus, while repair is improved in *srs2Δ rad55Δ* cells, it comes at the cost of very high level of crossover that is thought to drive genomic instability and loss of heterozygosity. Therefore, our data suggests that similar to what we demonstrated in BIR, the optimum Rad51 filament required for GC needs both Rad55 and Srs2 to be present. In addition, our data suggests that Srs2 is unlikely to fulfill its anti-crossover function by dislodging Rad51 from the non-invading DSB end[39] since this model predicted that *rad55Δ* would likely decrease, not increase, the crossover frequency in *srs2Δ*. Therefore, our results favor the model that Srs2 D-loop unwinding activity can account for its anti-crossover activity[36, 56]. As previously reported in a different system, the anti-crossover activity of Srs2 is largely dependent on its recruitment to SUMO-modified PCNA[25]. Consistently, in this ectopic recombination assay, the level of crossing-over is increased in mutants affecting SUMO-modification of PCNA, including *siz1Δ*, *pol30-K127R*, *pol30-K164R*, and the *pol30-K127R*,

*K164R* double mutant (Fig. 5c). Importantly, neither these mutants, nor *siz1Δ srs2-BRCΔ* had significant effect on viability (Fig. 5b), indicating that recruitment of Srs2 to recombination intermediates to promote efficient repair and to suppress crossover pathway are distinct. We also note that *siz1Δ* did not affect cell viability and the level of BIR (Fig. 3b, c), which is expected considering that disruption of the D-loop would rather counteract than help the D-loop migration. In addition, *siz1Δ srs2-BRCΔ* did not affect cell viability and the level of BIR as well, consistent with PCNA-independent recruitment of Srs2 to toxic intermediates.

We further hypothesized that the absence of the detox function of Srs2 was also responsible for the death that was previously observed in *srs2Δ* following DSB repair by SSA. In that system, DSB was introduced by HO endonuclease into the *LEU2* gene on chromosome III and could be repaired by SSA with a direct, partial *LEU2* sequence (*U2*) located 25 kb away[41] (Fig. 5d). More recently, it has been proposed that these DSBs could also be repaired by BIR via strand invasion of *LEU2* into *U2*[41]. The induction of this repair in *srs2Δ* resulted in loss of viability in 98% of cells[41] (Fig. 5e). Nevertheless, the authors observed an efficient formation of the repair product 6 h after the DSB detected by Southern blot analysis of the genomic DNA following its restriction digest and separation by gel electrophoresis[41]. We confirmed this finding in our experiments using Acc65I digested genomic DNA obtained from YMV80 and YMV88 (Supplementary Fig. 8). However, when we analyzed the repair in the same cells using CHEF gel electrophoresis, we observed no repaired chromosomes in YMV88 even after 12 h following DSB induction (Fig. 5f). We propose that despite the initiation of repair, the intact full chromosomes are never formed in *srs2Δ* cells. Thus, we propose that Srs2 plays an important detox role during repair in this SSA/BIR system as well. In addition, *srs2-BRCΔ* did not affect cell viability following DSB (Fig. 5e), indicative of the important role of Srs2 motor activity for survival.

## Discussion

BIR proceeds by an unusual, migrating bubble DNA synthesis, with conservative inheritance of newly synthesized strands[10, 11], and with asynchrony between the leading and lagging strand synthesis[10]. The latter leads to the formation of large amounts of ssDNA that is bound by RPA for protection, and can also bind Rad51. Our results suggest that unrestricted binding of Rad51 to ssDNA during BIR promotes unscheduled pairing to the homologous chromosome, which leads to the formation of toxic joint molecules that impede BIR and are lethal to the cell. We propose that Srs2 protects cells from these intermediates. According to our model (Fig. 6), the strippase activity of Srs2, which requires the ATPase and Rad51 interaction domains eliminates the excess of Rad51 from ssDNA regions formed during BIR (Fig. 6a, b), and this prevents the formation of toxic joint molecules. However, if they are nevertheless formed, Srs2 helicase activity, requiring ATPase domain, disrupts them (Fig. 6c). The toxic joint molecules formed in the absence of Srs2, can be resolved by Mus81-Mms4 or Yen1 (Fig. 6b), as we documented by showing: (i) higher viability in *srs2Δ* cells possessing Mus81 or Yen1[ON], and (ii) the change of the pattern of the branched DNA structures from the "spike" structures in *mus81Δ srs2Δ* to the 'rubble' in *srs2Δ* and in *srs2Δ mus81Δ YEN1[ON]*. We envision that the spike represents high molecular weight and highly branched intermediate that corresponds to multi-invasion regions formed behind the bubble (Supplementary Fig. 5). The resolution of some of the branched structures located inside the multi-invasion region by Mus81 and Yen1 leads to the formation of various molecules comprising the rubble intermediates. These multi-invasion intermediates are still branched because they include varying numbers of 4-way and 3-way junctions that still remain unprocessed (see Supplementary Fig. 5 and EM images in Supplementary Figs. 2, 3, and 6), which leads to varying complexity and varying molecular weight of these molecules comprising the rubble intermediates. These structures could also be dynamic and can be either processed further yielding linear molecules and survival, or may participate in re-formation of toxic structures resulting in cell death observed in 70% of the cases. We propose that the toxicity observed in *srs2Δ* mutants result from unprocessed 3-way junctions, 4-way junctions or other aberrant recombination intermediates formed by the ssDNA accumulated during BIR.

The viable BIR outcomes formed in *srs2Δ* via Mus81 resolution resemble those in wild-type cells, but are less frequently associated with mutations. This could result from lesser amounts of exposed ssDNA that can accumulate damage in *srs2Δ* as compared to wild-type cells due to a significant amount of ssDNA involved in the formation of toxic joint intermediates. Alternatively, it is possible that the mutagenic fraction of BIR is preferentially killed in *srs2Δ* mutants because it might form more complex multi-invasion structures. It is also possible that other aspects of BIR progression were affected in BIR survivors arising in the absence of Srs2. Overall, we report an important new risk for the cell that results from accumulation of ssDNA during BIR: it can be channeled into the formation of toxic joint molecules, and the goal of Srs2 is to prevent it from happening.

We propose that the role of Srs2 in removing toxic joint molecules is especially important in diploids where ssDNA regions located behind the D-loop share large regions of homology with homologous chromosomes. Moreover, our data suggests that a similar role might be played by Srs2 even in haploid cells, for example during SSA/BIR between distant repeats in YMV88 or ectopic gene conversion, where long ssDNA regions are formed[41, 46], but the areas for promiscuous invasion might be more limited. In particular, we observed that repaired chromosomes were virtually undetectable following SSA/BIR in YMV88 despite the successful formation of the initial repair fragments[41]. The failure of the repaired chromosomes to enter the CHEF gel is likely indicative of branched toxic joint molecules that can be formed by ssDNA regions generated by resection that proceeded outside of the annealing region in the case of SSA. These toxic intermediates could be formed by the invasion of ssDNA at ectopic positions, at locations of the Ty or delta elements, which can explain a low viability following DSB induction in haploid cells. In fact, using the same SSA system, it was demonstrated that long ssDNA region formed in a course of DSB resection contains a delta element that can invade at ectopic positions which modestly decreased cell viability even in the presence of Srs2[57]. It is likely that the absence of Srs2 can significantly exacerbate this problem. The importance of Srs2, and especially of its helicase activity for ectopic GC, as well as the restoration of cell viability in *srs2Δ* by deleting *RAD55* are indicative of an important anti-toxic role of Srs2 during ectopic GC as well. Therefore, we propose that Srs2 is required for the disruption of toxic joint molecules during ectopic GC and SSA, even though we cannot exclude its importance for other aspects of these DSB repair pathways, including successful DNA synthesis[44] or homology search[58]. In addition, while the DSB repair by BIR and ectopic GC is greatly improved in *srs2Δ rad55Δ* as compared to *srs2Δ* or *rad55Δ*, it remains significantly compromised. Our results suggest that Rad51 filament formed in the presence of Rad55/Rad57 and Srs2 proteins that stabilize and disrupt the nucleofilament, respectively, ensures optimal DSB repair kinetics, efficiency and prevents crossover outcomes.

Our study describes the case where successful repair of a DSB by recombination depends on successful elimination of aberrant recombination intermediates by Srs2. This function is similar to the "anti-recombination" function that Srs2 was previously proposed to play during S-phase replication[14–19, 59]. In particular, multiple studies have demonstrated that Srs2 prevents channeling of ssDNA gaps resulting from spontaneous and induced DNA damage into recombination that can be detrimental to the cell. The main difference though is that "ssDNA gaps" that can be formed during BIR are potentially much bigger as compared to those arising spontaneously, or in other DNA damage situations. Our idea that Srs2 plays the same role during BIR and broadly in DNA damage repair is supported by our observation that Yen1[ON] suppresses both BIR deficiency and MMS sensitivity of srs2Δ. Overall, based on our study we propose a comprehensive, yet simple, model where the same anti-recombination function of Srs2 is required for successful DSB repair, as well as to oppose spontaneous recombination.

Based on the detox role of Srs2 in BIR we predict that Srs2 is also important for the alternative lengthening of telomeres (ALT), a BIR-like pathway responsible for telomere maintenance in 10–15% of cancers[9]. We predict that the absence of Srs2 can result in trapping of chromosomes participating in ALT. Therefore, human counterparts of Srs2 may serve as anti-cancer targets in ALT cells. Overall, we propose that long ssDNA formed during BIR can form lethal joint molecules and we envision that this property can be used in the future development of anti-cancer therapy that will specifically target cells undergoing BIR.

## Methods

**Yeast strains and growth conditions**. All yeast strains were isogenic to AM1003[45], which is disomic for Chr. III with a genotype as follows: hmlΔ::ADE1/ hmlΔ::ADE3 MATa-LEU2-tel/MATα-inc hmrΔ::HPH FS2Δ::NAT/FS2 leu2/leu2-3112 thr4 ura3-52 ade3::GAL::HO ade1 met13.

AM3110 was derived from AM1003 in two steps. First, multiple copies of TEF1/ BSD were inserted into SNT1 of the MATα-inc containing copy of Chr. III using standard methods. Integration of multiple copies in Chr. III was confirmed by CHEF gel electrophoresis[10]. Second, p304-BrdU cassette was integrated into TRP1 on Chr. IV and the integration was confirmed by PCR[10]. All strains used in BIR assay were derivatives of AM3110. Strains used for the ectopic GC assays are derivatives of tGI354[40]. Strains used in SSA assay were derivatives of YMV80[41]. Derivatives containing srs2-BRCΔ were created using delitto perfetto approach where the pCORE plasmid[60] was integrated into the BRC region of SRS2 locus on Chr. X using 50 bp flanking homology on both sides. This cassette was replaced using 100 bp primers that are complementary to each other and are homologous to the sequence flanking SRS2 on both sides of the BRC region. This deletion was confirmed by sequencing (Supplementary Table 3). Strains containing srs2:: KANMX, pol30-K127R::KANMX, pol30-K164R::KANMX and siz1::KANMX were constructed using standard methods with PCR-derived KANMX module[41]. Strains containing mus81::ble[r], srs2::ble[r] and rad55::ble[r] were constructed using standard methods with PCR-derived phleomycin-resistant (ble[r]) cassette[61] flanked by terminal sequences matching the first and last 80 bp of the open reading frames of MUS81, SRS2 and RAD55 gene respectively. pol30-K127R::KANMX and pol30-K164R::KANMX strains were constructed by using standard methods with PCR-derived fragments amplified using genomic DNA of corresponding mutant strains[62].

Growth media contained rich medium yeast extract—peptone—dextrose (YEPD) and synthetic medium specific bases and amino-acids omitted as specified[46, 47]. YEP-lactate (YEP-Lac) and YEP-galactose (YEP-Gal) contained 1% yeast extract and 2% Bacto peptone media supplemented with 3.7% lactic acid (pH 5.5) or 2% (w/v) galactose, respectively. Cultures were grown at 30 °C.

**Physical analysis of DSB repair**. The kinetics and efficiency of BIR, ectopic GC and SSA was analyzed by CHEF gel electrophoresis followed by Southern blotting using 10× saline-sodium citrate (SSC) buffer and Southern hybridization using the following radiolabeled probes: ADE1 and ADE3-specific probes for the analysis of BIR repair product and donor Chr. III, respectively[10] and ADE1-specific probes for the analysis of Chr. III repaired by SSA[45]. Images were analyzed using a GE typhoon FLA 7000. Crossover frequency was measured by isolating DNA from cells 8 h and/or 24 h after induction of HO endonuclease. Isolated DNA was digested with EcoR1 and separated on a 0.8% agarose gel followed by Southern blotting and hybridization with radiolabeled probes. A 800 bp MATa fragment was

used as a probe[40]. Density of the GC and crossover bands was calculated using Bio-Rad Quantity One software. Two-dimensional (2D) gel electrophoresis analysis was done by extracting genomic DNA following cesium chloride density gradient centrifugation. Prior to 2D electrophoresis, the genomic DNA was digested by KpnI to analyze BIR intermediates at 0 kb, and by BglII for 24 kb and 85 kb positions. Digested DNA was first separated on a 0.4% agarose gel at 55 V (1D) and then separated on a 1.2% agarose gel at 340 V (2D)[10]. Following Southern blotting the intermediates at 0 kb, 24 kb, and 85 kb positions were detected by hybridizing with TAF2, PWP2, and CDC39 -specific radiolabeled probes, respectively.

**Genetic analysis of DSB repair**. Cell viability following DSB induction was determined by plating cells on YEPD and YEP-Gal media and calculated by dividing the number of colony-forming units (CFUs) on YEP-Gal by the number of CFUs on YEPD. A minimum of three plating experiments was performed to calculate the averages and standard deviations for viability. To characterize the DSB repair outcomes, the colonies formed on YEP-Gal plates were replica plated onto appropriate omission media to determine the fraction of DSB repair events with the following phenotypes: Ade[+] Leu[+] (GC), Ade[-white]Leu[−] (HC), Ade[-red] Leu[−] (CL), and Ade[+] Leu[−] (BIR)[45]. The rate of Lys[+] mutagenesis was determined by plating appropriate concentrations on YEPD media and on media omitting lysine before (0 h) and 7 h after the addition of galactose (7 h)[47]. The rate of Ura[+] mutagenesis was determined similar to the Lys[+] mutagenesis, except that appropriate concentration of cells were plated on YEPD media and on media omitting uracil before (0 h) and after (7 h) DSB induction[10]. The rate of mutations after galactose treatment was determined using a simplified version of the Drake equation. This modification was necessary because experimental strains did not divide or underwent one division between 0 h and 7 h[47].

**Analysis of BIR intermediates by electron microscopy**. EM analysis of BIR intermediates was by collecting samples 7 h following DSB induction and cross-linked using psoralen to preserve branched intermediates[10]. For No-DSB control, 0.015 mg/ml nocodazole was added simultaneously with 2% galactose, and the G2-arrested cells were collected 4 h later. The genomic DNA was extracted and processed similar to the samples in 2D gel electrophoresis[10], and digested with BglII. Following enrichment using BND cellulose[63], EM samples were prepared by spreading the DNA on carbon-coated grids in the presence of benzyl-dimethyl-alkylammonium chloride and visualized by platinum rotary shadowing[63]. Images were acquired on a transmission electron microscope (JEOL 1200 EX) with side-mounted camera (AMTXR41 supported by AMT software v601) and analyzed with ImageJ (National Institutes of Health).

**Data Availability**. All relevant data are available from the authors upon request.

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

## Acknowledgements

We thank Dr. Miguel Blanco for providing *YEN1^ON* plasmid. We thank Dr. Hannah Klein for providing *srs2-K41A* construct. We are grateful to Dr. Stefan Jentsch for the gift of *pol30* mutant strains. We thank Drs. James Haber, Wolf-Dietrich Heyer, and Michael Lichten for helpful discussions. The work was funded by NIH grants R01GM084242 to A.M., GM080600 and CPRIT RP140456 to G.I., NIH grant R01GM108648 and by DOD BRCP Breakthrough Award BC151728 to A.V.

## Author contributions

R.E., Z.S., K.L, G.I, A.V., and A.M. designed the experiments. R.E., J.D., N.P., D.L. constructed strains. R.E., J.D., C.S., Y.I., N.P., D.L., G.I. performed the BIR, ectopic GC and SSA experiments. Z.S., R.E., and K.L. performed 2D gel electrophoresis. J.J., R.E., A.

V. performed EM experiments. R.E., G.I, and A.M. wrote the manuscript. R.E. and Z.S. contributed equally to this work.

## Additional information

**Competing interests:** The authors declare no competing financial interests.

