## [Peer Review File · Nature Communications]

Reviewers' comments:

Reviewer #1 (Remarks to the Author):

The manuscript by Elango et al focuses on the roles of Srs2 as a “strippase” that unloads the recombinase Rad51 from ssDNA and as a DNA helicase in preventing the formation of toxic DNA joint molecule intermediates. Using break-induced replication (BIR), crossover, and single-strand annealing (SSA) assays, the authors have addressed the importance of Srs2 in dislodging Rad51 and removing toxic DNA intermediates. Their analyses furnish evidence that the Rad51-strippase activity of Srs2 prevents the formation of toxic intermediates and its helicase activity removes the DNA intermediates. They also demonstrate that the nucleases Mus81 and Yen1 are responsible for resolving toxic joint molecules formed in the absence of Srs2 in BIR. In addition, the finding that perturbation of the Rad51 presynaptic filament, i.e. in the *rad55Δ* or hypomorphic RPA (*rfa1-t33*) mutant, helps alleviate the *srs2Δ* phenotype in BIR and crossover.

This manuscript addresses novel aspects of Srs2 and the results are convincing and support the conclusions drawn. Overall, this is a very nice story that will have a significant impact on understanding the regulation of recombination pathways and the maintenance of genomic stability in eukaryotes.

I have just a few textual points to consider.

Specific Comments:

(1) The strippase activity of *srs2-ΔBRC* is less than Srs2 WT but not null as shown by Colavito et al 2009 and Antony et al 2009, especially when the Rad51 filament is short. In this regard, there could be residual Rad51 removal from the presynaptic filament by Srs2 in an interaction independent manner and the interpretation of the *srs2-ΔBRC* data needs to consider this aspect. In particular, the phenotypic difference between *srs2-ΔBRC* and *srs-K41A* may not stem from Srs2 unwinding of DNA intermediates solely.

(2) Line 326: the authors state that: “Similarly, in this ectopic recombination assay anti-crossover function of Srs2 nearly entirely depends on Siz1 SUMO ligase and Pol30 K127, K164 sumoylation as analyzed in *siz1Δ*, *pol30-K127R*, *pol30-K164R*, and the *pol30-K127R*, *K164R* double mutant.” The results in Figure 5 are on the *pol30* SUMO mutants but not in conjunction with *srs2Δ*. Please consider rewriting the passage in question.

(3) The authors call the toxic DNA intermediates “spike” in *srs2Δ mus81Δ* and “rubble” in *srs2Δ*. Please explain what the differences are.

(4) Line 384: ‘slower progression of BIR bubble’... How do the authors envision *srs2Δ* or toxic intermediates in ssDNA cause BIR bubble migration to slow?

Reviewer #2 (Remarks to the Author):

The manuscript by Elango et al. examines the role of the Srs2 helicase of yeast *Saccharomyces cerevisiae* in the repair reaction known as “break-induced replication” (BIR). BIR is of particular interest because it constitutes a mechanism that can repair broken replication forks and contributes to telomere maintenance in an alternative pathway to telomerase (the “ALT” pathway). BIR can also mediate one-ended double-strand break repair with consequent loss of heterozygosity

for the distal portion of the chromosome. DNA synthesis during BIR is distinctly different from that in S phase, which Malkova and collaborators have shown occurs by conservative synthesis via "bubble migration".

Using a number of genetic assays for BIR (and mutagenesis and homologous recombination) as well as electrophoretic DNA analysis, the authors demonstrate that BIR requires Srs2. In *srs2* mutants BIR is associated with cell lethality, correlated with a loss of BIR product detected in pulse-field gels and an aberrant, slowly migrating species on Brewer-Fangman gels (that the authors term "rubble"). BIR-associated hypermutation is also lowered in *srs2* mutants. Loss of structure-specific endonuclease Mus81 exacerbates toxicity of BIR without Srs2 whereas mutations that allow an alternative endonuclease Yen1 to be expressed outside of M phase can suppress *srs2* phenotypes. The authors demonstrate that the ATPase of Srs2 is necessary for promotion of BIR; loss of the Rad51 interaction domain of Srs2 has weaker effects. Previously, the Siz1 SUMO ligase and sumoylation of the PCNA clamp have been shown to be required for Srs2's role in suppressing crossover recombination. Interestingly, these are not required for Srs2's role in avoiding toxic BIR. Finally, *rad55* mutations, which destabilize the Rad51 filament, also suppress BIR toxicity. The authors present a model whereby ssDNA that accumulates during BIR initiates toxic recombination events that are aborted by dissolution via Srs2.

The strength of the manuscript is that through the genetic analysis a convincing story is built for a "detox" role for Srs2 in BIR. That this toxic intermediate is branched DNA is supported by the genetic effect of Holliday junction resolution enzymes and by the appearance of new structures in the 2D gels. Interpretation of the 2D gels is the weakest part of the manuscript and involves a leap of faith on the part of the reader--the authors show a very limited example of 3 and 4-strand branches observed by electron microscopy. I suspect that the BND cellulose enrichment is not quantitative, making the 2-fold effect of *srs2* in Fig. 2b meaningless (in addition, EM-based scoring can be very prone to bias). Whether unresolved branches constitute all the toxic intermediates is unclear to me. That recombinant molecules are subject to degradation and loss or fail to be completely replicated seems like some other possibilities.

There are a number of wording problems that make the manuscript confusing at points and the conclusions are a bit overstated. An antirecombination role for Srs2 (and other helicases) has been known for some time and the authors should be thorough in their citation of this in the discussion.

line 50 "instabilities" should be "instability"

line 68 should be "the BIR defect"

line 96 please explain the unloading hypothesis more if you're going to reinterpret these data

line 104 reword" in the absence of Srs2, the structure-specific endonucleases Mus81 and Yen1 can resolve"

line 120 "and is followed by strand invasion"

line 267 another striking result is that Srs2 promotes chromosome loss in *rad55* mutants, at expense of BIR, this should be noted

line 281 "Additionally, the *srs2-BCRAΔ* allele restored BIR in *rad55Δ* mutants"

line 287 "a defect"

line 349 this point needs some more elaboration since SSA doesn't involve any HJ or D-loops; are the authors suggesting that ssDNA during SSA initiates ectopic pairing? Does the resected region include Ty and other repetitive elements?

Apropos to this point, it was proposed that UvrD (arguably the bacterial Srs2 equivalent), acts as a recombination "proofreader" to abort recombination at short or weakly homologous sequences (Morel et al. 1993 NAR 21: 3205), something that should be cited.

line 359 "demonstrates" seems too strong a term here, since the authors do not measure binding of Rad51 in any way. Better to say "is consistent with the hypothesis" or something like it. This whole paragraph should be more circumspect, replacing "we report" with "what we observe is

consistent with"

line 380 do the authors mean nicked junctions? I'm not sure what other intermediate products would be observed? Why would they migrate slower? This section needs more development

line 413 not seeing the product on the CHEF gel is not necessarily indicative of branched toxic joint molecules; the product could be degraded, or covalently linked to proteins, for example. Please be less strong here.

line 429 it seems to me that the "anti-recombinogenic function" could be a "maturation function" ; please be more clear

Fig. 1a and b: This is confusing as the molecular model diagrams appear to read left to right; please sequester part b so that it doesn't look like part of the a models.

Reviewer #3 (Remarks to the Author):

Elango and colleagues report that SRS2 plays an important role in BIR by removing toxic intermediates that have the potential to disrupt this process. They propose that SRS2 achieves this by removing RAD51 from ssDNA, thereby preventing unwanted strand exchange reactions that hinder BIR and also by helping remove toxic intermediates that form behind the replication bubble.

This is a robust study of the genetic contribution of SRS2 to BIR. The data are well presented and the authors' arguments are easy to follow. Genetically, the experiments seem pretty solid to someone who is not a yeast geneticist. Where I have more difficulty is the inference of mechanism solely through genetic dissection. Yes, the data might be consistent with the model presented but no attempt is made to use cell biological or biochemical methods to probe the model beyond using standard 2D gels. In similar studies on repair/recombination pathways in human cells the authors would be expected to use an array of cellular and biochemical techniques to identify the localization and kinetics of proteins involved in such a process. As a result, the approach and findings of the work here come across as esoteric and based on a considerable amount of inference.

Response to reviewer's comments.

This letter contains our point-by-point responses to the comments of the referees. Several comments (including comments made by Reviewer 2 concerning the absence of SSA repair products and intermediates on CHEF gel) have been addressed experimentally. Several additional Supplementary figures were created in response to the comments from Reviewers 1 and 2. These Supplementary figures include additional EM images illustrating different types of DNA repair intermediates formed in the absence of Srs2. In addition, our new Supplementary Fig. 5 includes schematics explaining the proposed structures of the rubble and spike intermediates, as well as our models explaining their formation. Also, in response to the reviewer's comments that suggested alterations to the text and figures, we either made these changes as requested or discussed them in detail. Finally, the last part of the letter addresses the concerns of the Reviewer 3 regarding the choice of methods that were used in our study. In addition, we shortened the abstract to conform to the requirements of this journal. All the textual changes that were made to the manuscript are highlighted in our revised manuscript. It is our hope, that the revisions made to our manuscript are satisfactory and that our paper is now suitable for publication in Nature Communications.

Detailed answers to specific comments from Reviewer 1

1. **Reviewer 1 states:** “This manuscript addresses novel aspects of Srs2 and the results are convincing and support the conclusions drawn. Overall, this is a very nice story that will have a significant impact on understanding the regulation of recombination pathways and the maintenance of genomic stability in eukaryotes.”

Answer: We thank this reviewer for considering our work “convincing” and in “support of the conclusions drawn”

2. **Reviewer 1 states:** “The strippase activity of *srs2-ΔBRC* is less than Srs2 WT but not null as shown by Colavito et al 2009 and Antony et al 2009, especially when the Rad51 filament is short. In this regard, there could be residual Rad51 removal from the presynaptic filament by Srs2 in an interaction independent manner and the interpretation of the *srs2-ΔBRC* data needs to consider this aspect. In particular, the phenotypic difference between *srs2-ΔBRC* and *srs-K41A* may not stem from Srs2 unwinding of DNA intermediates solely”

Answer: We thank the reviewer for this question and would like to point out that we did take into account the possibility that the strippase activity of *srs2-ΔBRC* might not be null. In fact, while choosing the Srs2 mutation for our study, we noticed that several published *srs2* alleles containing deletions of Rad51-interaction domain showed a variation in their residual strippase activities. Specifically, the paper by Colavito et al.¹ mentioned by the reviewer described several such mutants. The main mutant that has been used extensively in this paper was *srs2-Δ875-902*. In a number of *in vitro* tests this mutation led to severe defect in interaction between Srs2 and Rad51 as well as to defective strippase activity. In

particular, this mutant protein was very defective in suppressing the strand exchange reaction as well as in disruption of Rad51 filament as measured by electron microscopy. The leftover recombination activity observed in this mutant was very low. The caveat of this mutation was that it failed to suppress MMS sensitivity of *rad18Δ* *in vivo* that led to speculations about its leakiness. The Antony et al paper² used the *Srs2*^{CA314} mutation, which is similar to *srs2-860*¹. This mutation led to a significant defect in strippase activity in their *in vitro* experiments as more than 80% of Rad51 remained bound to the filament in the presence of this mutant protein while all of the Rad51 was removed by a wild-type protein. This mutation completely eliminates the whole Rad51-interaction domain of *SRS2*, but in addition it also eliminates other domains, including the PCNA-interacting domain, and a number of Srs2 phosphorylation sites, which are critical for the regulation of Srs2 in response to DNA damage. In addition, it has been recently demonstrated that this *srs2-860* mutation leads to two-fold increase of Srs2 expression³ which results in unusual and residual activities of this protein. This is why we decided not to use this mutation, even though in our preliminary experiments it did actually show the phenotypes similar to those we observed in the *srs2-BRCΔ* used in our study. The *srs2-BRCΔ* that we used in our experiments contained the deletion of the Srs2 domain from 862 to 914. This deletion is bigger than *srs2-Δ875-902*, used in Colavito et al. and includes the entire Rad51 binding domain of Srs2. Importantly, *srs2-Δ862-914* did suppress the MMS sensitivity of *rad18Δ* in our strain background, which was an important advantage over *srs2-Δ875-902*. In addition, this mutant protein contained all of the other interaction and regulation sites of Srs2, and therefore was unlikely to exhibit any unusual and de-novo effects.

In addition, we would like to point out that in our experiments on BIR, we observed multi-invasion structures (referred as a “rubble”) in the *srs2-Δ862-914* mutant. This suggests that even if some residual strippase activity is still present in these mutants, it could not prevent the formation of multi-invasion structure (see Fig. 3d and 3g). Once these structures are formed, the strippase activity of Srs2 is unlikely to unwind them, but will require the motor activity of Srs2, likely its helicase activity as shown in previous *in vitro* experiments^{4,5}. Therefore, we believe that the helicase activity of Srs2 is the most valid candidate for the unwinding of the toxic intermediates once they are formed. However, in the case of GC and SSA while analyzing effects of *srs2-BRCΔ*, we relied solely on genetic results, thus we modified our conclusion according to suggestions from this reviewer (Page 14 and 16, line 292 and 337 respectively).

Overall in response to this reviewer’s comment we have corrected the text in the following way.

Page 2. Line 28. Here, we propose that uncontrolled Rad51 binding to this ssDNA promotes formation of toxic joint molecules that are counteracted by Srs2. First, Srs2 dislodges Rad51 from ssDNA preventing promiscuous strand invasions. Second, it dismantles toxic intermediates that have already formed.

Page 5. Line 94. We propose that two main activities of Srs2, strippase and helicase, counteract toxic joint intermediates by preventing their formation and promoting their disruption, respectively.

Page 11. Line 240. Based on these results we propose that the defect of *srs2-BRCΔ*, in removal of the Rad51 from the filament¹ leads to the formation of toxic intermediates that are eventually removed by the helicase activity of Srs2, allowing slower but efficient repair and survival.

Page 11. Line 246. Unstable Rad51 filament bypasses the need for Srs2.

Page 14. Line 292. In addition, we observed that *srs2-BRCΔ* did not affect cell viability during GC (Fig. 5b), indicative of the important role of Srs2 motor activity for survival.

Page 16. Line 337. In addition, *srs2-BRCΔ* did not affect cell viability following DSB (Fig. 5e), indicative of the important role that Srs2 motor plays in survival.

3. Reviewer 1 states: Line 326: the authors state that: “Similarly, in this ectopic recombination assay anti-crossover function of Srs2 nearly entirely depends on Siz1 SUMO ligase and Pol30 K127, K164 sumoylation as analyzed in *siz1Δ*, *pol30-K127R*, *pol30-K164R*, and the *pol30-K127R, K164R* double mutant. The results in Figure 5 are on the *pol30* SUMO mutants but not in conjunction with *srs2Δ*. Please consider rewriting the passage in question.”

Answer: We re-wrote the sentence Page 14. Line 311 in the following way: Consistently, in this ectopic recombination assay, the level of crossing-over is increased in mutants affecting SUMO-modification of PCNA, including *siz1Δ*, *pol30-K127R*, *pol30-K164R*, and the *pol30-K127R, K164R* double mutant (Fig. 5c). Importantly, neither these mutants, nor *siz1Δ srs2-BRCΔ* had significant effect on viability (Fig. 5b), indicating that recruitment of Srs2 to recombination intermediates to promote efficient repair and to suppress crossover pathway are distinct.

4. Reviewer 1 states: “The authors call the toxic DNA intermediates “spike” in *srs2Δ* *mus81Δ* and “rubble” in *srs2Δ*. Please explain what the differences are.”

Answer: We propose that both unusual patterns observed during 2D gel electrophoresis which, we called ‘spike’ and ‘rubble’, are joint molecules resulting from complex strand invasions formed behind the BIR bubble in the absence of Srs2. ‘Spike’ is a product that corresponds to more uniform invasion structure observed in *srs2Δ* only in the absence of structure specific nucleases Mus81. Processing of this structure by Mus81 or Yen1^{ON} leads to much less uniform and diffused structure that we call rubble (Supplementary Fig. 5).

We propose that DNA species corresponding to ‘spike’ and ‘rubble’ structures migrate differently in agarose gel because of different complexity (‘branchiness’) and possibly molecular weight. A hypothetical model is proposed in Supplementary Fig. 5. In *srs2Δ*

mus81Δ only BglII sites located outside of this multi-invasion structure can be cut and this leads to the formation of high molecular weight species that are also highly branched. We envision that they consist of several 4-way and 3-way junction structures (see Supplementary Fig. 5 and Supplementary Fig. 6), and that processing of these junctions by structure-specific endonucleases, Mus81 and Yen1 (Supplementary Fig. 5, red and yellow arrowheads, respectively) leads to the formation of nicked intermediates that could also undergo resection (blue pacman) or unwinding. The resulting intermediates will have various levels of complexity and possibly reduced molecular weight and therefore migrate on a 2D gel in a more diffused way and below the position of the spike. The complexity of these intermediates will depend on the number of the remaining (unprocessed) 4-way and 3-way junctions inside them, and therefore these intermediates will form a heterogeneous pattern of migration on 2D gel that we call ‘rubble’.

In response to this reviewer’s comment the following changes have been introduced in the text of the paper:

Page 9. Line 184. “ We propose that these structures represent the high molecular weight and highly branched intermediate corresponding to the multi-invasion regions formed behind the bubble (See Supplementary Fig. 5, Supplementary Fig. 6 and discussion for details). We envision that partial resolution of these multi-invasion intermediates by Mus81 leads to formation of transient intermediates that are visualized as rubbles on 2D gel electrophoresis. These transient intermediates can be processed further, which allows formation of viable BIR outcomes.

Page 17. Line 357. ” We envision that the spike represents high molecular weight and highly branched intermediate that corresponds to multi-invasion regions formed behind the bubble (Supplementary Fig. 5). The resolution of some of the branched structures located inside the multi-invasion region by Mus81 and Yen1 leads to the formation of various molecules comprising the rubble intermediates. These multi-invasion are still branched because they include varying numbers of 4-way and 3-way junctions that remain unprocessed (see Supplementary Fig. 5 and EM images in Supplementary Fig. 2, 3 and 6), which leads to varying complexity and molecular weight of these molecules comprising the rubble intermediates. These structures are dynamic and can be either processed further yielding linear molecules and survival, or may participate in re-formation of toxic structures resulting in cell death observed in 70% of the cases. We propose that the toxicity observed in *srs2Δ* mutants result from unprocessed 3-way junctions, 4-way junctions or other aberrant recombination intermediates formed by ssDNA accumulated during BIR.

5. **Reviewer 1 states:** line 384: “slower progression of BIR bubble’.... How do the authors envision *srs2Δ* or toxic intermediates in ssDNA cause BIR bubble migration to slow? “

Answer: In response to this reviewer’s comment, we would like clarify that our idea of the reduced speed of BIR progression in *srs2Δ* mutants came from our observation of the reduced mutagenesis in *srs2Δ*. However, we agree that the reduced mutagenesis can be

explained by many other changes in the mechanism of BIR that take place in *srs2Δ* cells as well. The most likely explanation of the reduced mutagenesis is that lesser amounts of exposed ssDNA present in *srs2Δ* as compared to wild type cells. This could be because a significant portion of ssDNA is involved in the formation of toxic joint intermediates. The accumulation of damage in ssDNA is a potent source of base substitutions associated with BIR, but also can promote formation of frameshifts since a fraction of frameshifts were previously shown to be dependent on Polζ⁶. It is also possible that a mutagenic fraction of BIR events that accumulates a large amount of ssDNA is preferentially killed in *srs2Δ*. It is also possible that *srs2Δ* changes some other features of BIR synthesis contributing to mutagenesis for example, frequent dissociation of Polδ, inefficient mismatch repair or speed of bubble-migration⁶. Less frequent frameshifts can also result from processing of the intermediates by structure-specific endonucleases, or due to preferential death of cells forming pre-mutagenic intermediates. Overall the most important conclusion from our analysis of BIR associated mutagenesis is that BIR progression was affected in *srs2Δ* survivors. Therefore, in response to the reviewers concern, we changed our statement in the following way:

Page 17. Line 375. “Alternatively, it is possible that the mutagenic fraction of BIR is preferentially killed in *srs2Δ* mutants because it might form more complex multi-invasion structures. It is also possible that other aspects of BIR progression were affected in BIR survivors arising in the absence of Srs2”.

Detailed answers to specific comments from Reviewer 2:

1. Reviewer 2 states: “The strength of the manuscript is that through the genetic analysis a convincing story is built for a “detox” role for Srs2 in BIR. That this toxic intermediate is branched DNA is supported by the genetic effect of Holliday junction resolution enzymes and by the appearance of new structures in the 2D gels”

Answer: We thank the reviewer for this comment.

2. Reviewer 2 states: “Interpretation of the 2D gels is the weakest part of the manuscript and involves a leap of faith on the part of the reader--the authors show a very limited example of 3 and 4-strand branches observed by electron microscopy. I suspect that the BND cellulose enrichment is not quantitative, making the 2-fold effect of *srs2* in Fig. 2b meaningless (in addition, EM-based scoring can be very prone to bias). Whether unresolved branches constitute all the toxic intermediates is unclear to me. That recombinant molecules are subject to degradation and loss or fail to be completely replicated seems like some other possibilities”.

Answer: In this study, we described two new 2D structures, ‘spike’ and ‘rubble’. These are two types of joint molecules and both are observed only in *srs2Δ* cells, where ‘rubble’ results from the cleavage of multi-invasion intermediate by Mus81 or Yen1^{ON}. We proposed a model that could explain the exact nature of these structures in Supplementary Fig. 5. The model is based on known mechanism of BIR involving long ssDNA intermediates, the results of 2D gel electrophoresis, genetic analysis and electron

microscopy. We want to point out that unlike other recombination structures previously interpreted by others (single or double HJ) these do not correspond to a single spot but rather long diffused arcs and spikes pointing at lack of unique or uniform structure. Please, also see our response to Reviewer #1 for our interpretation of their structure and their formation.

Importantly, even following resolution with Mus81, the intermediates migrated above the line of linear DNA molecules (Fig. 1g, Supplementary Fig.5) suggesting that some 3-way and 4-way structures were still present in the rubble intermediate. This was confirmed by detection of 3-way and 4-way junctions by EM as documented in Fig. 2a, and in Supplementary Figs. 2, 3 and 6 where we included additional EM images, as requested by this reviewer. The main goal of the EM analysis was not quantitative but qualitative in nature: to test the presence of 3-way and 4-way junctions. In order to find these structures corresponding to intermediates of BIR, we had to use BND cellulose to enrich for recombination intermediates. This procedure was absolutely necessary in our experiments because only one chromosome participated in BIR and therefore it is almost impossible to detect BIR intermediates without enrichment. This approach is routinely used in the laboratory of Dr. Alessandro Vindigni, an author on this paper, as well as in the laboratory of Dr. Massimo Lopes, for the analysis of replication intermediates. Multiple publications from these two labs⁷⁻¹⁰ demonstrate that reliable quantitative conclusions can be drawn from experiments where both experimental and control DNA have been subjected in parallel to BND cellulose enrichment. Therefore, we believe that the conclusions derived from our work are statistically sound as well. Most importantly, we would like to emphasize that 4-way and 3-way junctions were observed only in samples where a DSB was induced and were not observed in no-cut controls (Supplementary Table 2). These no-cut samples served as an important internal control that helped us to validate the usage of BND cellulose technique. In addition, we observed that the amount of branched molecules was enriched in *srs2Δ* as compared to BIR samples. These conclusions were based on the results of 3 independent experiments where thousands of molecules were counted (Supplementary Table 2), and therefore we believe that our two-fold increase observed in *srs2Δ* is statistically sound.

In response to the reviewer's concern whether unresolved branches indeed constitute toxic intermediates, we would like to point out that this conclusion was made based on the important role of the presence of Mus81 and Yen1 for the viability of *srs2Δ* cells. Since branched intermediates are a known substrate of Mus81 and Yen1, it seems reasonable for us to propose that unresolved 3-way and 4-way junctions are responsible for the death of *srs2Δ* cells even though we cannot fully exclude that some other structures, for example broken or under-replicated recombination intermediates could contribute to this toxicity as well, as suggested by the reviewer.

In response to the Reviewer's comments, we have made the following changes in the text and figures:

1. A new schematic including the structure and individual steps that lead to the transitions between spike and rubble intermediates is provided in Supplementary Fig. 5. The detailed

interpretation of these structures and of corresponding 2D gel images is provided in the legend to this figure.

2. As suggested by the Reviewer 2, many additional EM images are included in Supplementary Fig. 2, 3 and 6. Importantly, our new classification of these images (Supplementary Fig. 6) helps to illustrate different intermediates that are presented in the schematic in Supplementary Fig. 5.

3. Page 17. Line 357. We envision that the spike represents high molecular weight and highly branched intermediate that corresponds to multi-invasion regions formed behind the bubble (Supplementary Fig. 5). The resolution of some of the branched structures by Mus81 and Yen1 located inside the multi-invasion region leads to the formation of various molecules comprising the rubble intermediates. These molecules are still branched because they include varying numbers of 4-way and 3-way junctions that still remain unprocessed (see Supplementary Fig. 5 and EM images in Supplementary Figs. 2, 3 and 6), which leads to varying complexity and varying molecular weight of these molecules comprising the rubble intermediates. These structures could also be dynamic and can be either processed further yielding linear molecules and survival, or may participate in re-formation of toxic structures resulting in cell death observed in 70% of the cases. We propose that the toxicity observed in *srs2Δ* mutants result from unprocessed 3-way junctions, 4-way junctions or other aberrant recombination intermediates formed by the ssDNA accumulated during BIR.

3. Reviewer 2 states: “There are a number of wording problems that make the manuscript confusing at points and the conclusions are a bit overstated”.

Answer: We apologize for this. We have addressed all wording concerns, as it is detailed below.

4. Reviewer 2 states: An antirecombination role for Srs2 (and other helicases) has been known for some time and the authors should be thorough in their citation of this in the discussion.

Answer: In response to this reviewer’s comment the following citations to the previous research on the anti-recombination role of Srs2 have been included in Page 3 Line 67:

In addition, a number of other genetic studies proposed an anti-recombination role played by Srs2¹¹⁻²⁶.

5. **Reviewer 2 states:** line 50 “instabilities” should be “instability”

Answer: This has been corrected.

6. **Reviewer 2 states:** line 68 should be “the BIR defect”

Answer: This has been corrected.

7. **Reviewer 2 states:** line 96 please explain the unloading hypothesis more if you're going to reinterpret these data

Answer: This comment has been addressed. The new sentence on Page 4 Line 85 now states: "Later, this death was attributed to the incomplete unloading of recombination factors leading to persistent binding of Rad51 and RPA to the ssDNA surrounding the areas of repair even after the repair has been completed^{27,28}. Together these studies proposed different pro-recombination roles of Srs2 in DSB repair."

8. **Reviewer 2 states:** line 104 reword "in the absence of Srs2, the structure-specific endonucleases Mus81 and Yen1 can resolve"

Answer: This has been corrected

9. **Reviewer 2 states:** line 120 "and is followed by strand invasion"

Answer: This has been corrected

10. **Reviewer 2 states:** line 267 "another striking result is that Srs2 promotes chromosome loss in rad55 mutants, at expense of BIR, this should be noted"

Answer: This comment has been addressed. The new sentence on Page 12 Line 251 now states: The latter observation suggests that Srs2-dependent removal of Rad51 in the absence of Rad55 prevents successful strand invasion, which leads to chromosome loss.

11. **Reviewer 2 states:** line 281 "Additionally, the *srs2-BRCA* allele restored BIR in rad55Δ mutants".

Answer: This has been done

12. **Reviewer 2 states:** line 287 "a defect"

Answer: This has been corrected

13. **Reviewer 2 states:** line 349 "this point needs some more elaboration since SSA doesn't involve any HJ or D-loops; are the authors suggesting that ssDNA during SSA initiates ectopic pairing? Does the resected region include Ty and other repetitive elements?"

Answer: There are at least two possible scenarios of how toxic intermediates could be formed in YMV88 in the absence of Srs2. First, it has been recently found^{27,29} that an HO-induced break initiated at *LEU2* can be repaired not only by SSA, but also by BIR. Our corrected version of the manuscript acknowledges this possibility.

In any case, our observation of important anti-toxic role of Srs2 in this SSA/BIR system is important since it allowed us to explain death of *srs2* Δ cells that was previously ascribed to unexplained checkpoint recovery defects³⁰.

Second, it is also possible that toxic intermediates result from ectopic invasion of ssDNA containing delta elements located in the region centromere distal to the *LEU2::HOcs* cassette³¹. We discuss this possibility in the text of the paper (Page 18 Line 391): “These toxic intermediates could be formed by the invasion of ssDNA at ectopic positions, at locations of the Ty or delta elements, which can explain a low viability following DSB induction in haploid cells. In fact, using the same SSA system, it was demonstrated that long ssDNA region formed in a course of DSB resection contains a delta element that can invade at ectopic positions which modestly decreased cell viability even in the presence of Srs2³¹.”

In response to these comments, the text has been edited as follows:

Page 15. Line 326. “More recently, it has been proposed that these DSBs could also be repaired by BIR via strand invasion of *LEU2* into *U2*. The induction of this repair in *srs2* Δ resulted in loss of viability in 98% of cells (Fig. 5e). Nevertheless, the authors observed an efficient formation of the repair product 6 h after the DSB detected by Southern blot analysis of the genomic DNA following its restriction digest and separation by gel electrophoresis³⁰. We confirmed this finding in our experiments using Acc65I digested genomic DNA obtained from YMV80 and YMV88 (Supplementary Fig. 8). However when we analyzed the repair in the same cells using CHEF gel electrophoresis we observed no repaired chromosomes in YMV88 even after 12 h following DSB induction (Fig. 5f). We propose that despite the initiation of repair³⁰, the intact full chromosomes are never formed in *srs2* Δ cells. Thus, we propose that Srs2 plays an important detox role during repair in this SSA/BIR as well. In addition, *srs2-BRC* Δ did not affect cell viability following DSB (Fig. 5e), indicative of the important role of Srs2 motor activity for survival.”

Page 18. Line 383. “Moreover, our data suggests that a similar role might be played by Srs2 even in haploid cells, for example during SSA/BIR between distant repeats in YMV88 or ectopic gene conversion, where long ssDNA regions are formed^{30,32}, but the areas for promiscuous invasion might be more limited. In particular, we observed that repaired chromosomes were virtually undetectable following SSA/BIR in YMV88 despite the successful formation of the initial repair fragments³⁰.”

14. **Reviewer 2 states:** Apropos to this point, it was proposed that UvrD (arguably the bacterial Srs2 equivalent), acts as a recombination “proofreader” to abort recombination at short or weakly homologous sequences (Morel et al. 1993 NAR 21: 3205), something that should be cited.

Answer: Thank you for bringing this to our attention. To address this comment, the following statement has been included in the text:

Page 3 Line 67. In addition, a number of other genetic studies proposed an anti-recombination role played by Srs2¹¹⁻²⁶ and also by its bacterial functional homolog, UvrD³³.

15. **Reviewer 2 states:** line 359 “demonstrates” seems too strong a term here, since the authors do not measure binding of Rad51 in any way. Better to say “is consistent with the hypothesis” or something like it. This whole paragraph should be more circumspect, replacing “we report” with “what we observe is consistent with”

Answer: To address this comment, the corresponding paragraph has been re-written in the following way:

Page 16. Line 344. “Our results suggest that unrestricted binding of Rad51 to ssDNA during BIR promotes unscheduled pairing to homologous chromosome, which leads to the formation of toxic joint molecules that impede BIR and are lethal to the cell. We propose that Srs2 protects cells from these intermediates.”

16. **Reviewer 2 states:** line 380 “do the authors mean nicked junctions? I’m not sure what other intermediate products would be observed? Why would they migrate slower? This section needs more development”

Answer: In response to this reviewers comment, we would like to clarify that according to our model (see schematic in Supplementary Fig. 5), the multi-invasion structures forming the ‘spike’ contain several 3-way and 4-way junctions. We also propose that the formation of the rubble results from the resolution of only some of these junctions by Yen1 and Mus81 while other junctions remain unresolved and are present inside large nicked intermediates. Consistently, the presence of these structures is confirmed by our EM analysis of *srs2Δ* (Fig. 2, Supplementary Fig. 2, 3 and 6). We believe that these structures are responsible for the slower migration of the rubble intermediates above the line of linear molecules (see our responses #4 to Reviewer 1 and #1 to Reviewer 2 and also refer to Supplementary Fig. 5 for more details).

17. **Reviewer 2 states:** line 413 “not seeing the product on the CHEF gel is not necessarily indicative of branched toxic joint molecules; the product could be degraded, or covalently linked to proteins, for example. Please be less strong here.”

Answer: We agree with the reviewer that the absence of DNA on CHEF alone does not necessarily mean accumulation of branched intermediate. To make sure that DNA is not degraded we performed additional experiments. First, we confirmed that similar to what was previously reported, the product of repair following DSB could be detected by digesting genomic DNA by Acc65I followed by hybridization with *LEU2* probe (Supplementary Fig. 8). This product of repair was readily detected in both wild-type and *srs2Δ* and therefore was not degraded suggesting that the corresponding chromosome undergoing repair was not degraded as well. Second, to make sure that the product of DNA repair was not degraded during preparation of agarose plugs for CHEF analysis, we used the same Acc65I restriction enzyme and treated the agarose plugs containing

chromosomal DNA in two different ways. First, we treated the intact plugs with Acc65I enzyme and allowed for in-plug digestion (see new Supplementary Fig. 8b). Second, we extracted chromosomal DNA from the plugs using beta-agarase enzyme and then digested it with Acc65I (Supplementary Fig. 8c). Following these two methods of plug digestion, we separated the fragments on a gel and hybridized with *LEU2* probe. We observed the formation of a band indicative of intact repair product using both methods consistent with our prior conclusion that the chromosome undergoing repair is not degraded. These structures were also unlikely to be covalently cross-linked with proteins since we pre-treated the agarose plugs with Proteinase K that should eliminate all proteins in the sample. We conclude that the repaired DNA product formed following DSBs is not degraded and fails to enter the CHEF gel likely due to its branched nature. Overall in response to this comment, we included new data in Supplementary Fig. 8. In addition, the following corrections were made to the text:

Page 15. Line 331. We confirmed this finding in our experiments using Acc65I digested genomic DNA obtained from YMV80 and YMV88 (Supplementary Fig. 8). However, when we analyzed the repair in the same cells using CHEF gel electrophoresis we observed no repaired chromosomes even after 12 h following DSB induction (Fig. 5f).

Page 18. Line 388. “The failure of the repaired chromosomes to enter the CHEF gel is like indicative of branched toxic joint molecules...”

18. **Reviewer 2 states:** line 429 “it seems to me that the “anti-recombinogenic function” could be a “maturation function”; please be more clear”

Answer: In response to this comment, the following change was made to the text:

Page 19. Line 410. Our study describes the case where successful repair of a DSB by recombination depends on successful elimination of aberrant recombination intermediates by Srs2. This function is similar to the “anti-recombination” function that Srs2 was previously proposed to play during S-phase replication^{11,22-24,34-36}.

19. **Reviewer 2 states:** Fig. 1a and b: This is confusing as the molecular model diagrams appear to read left to right; please sequester part b so that it doesn’t look like part of the a models.

Answer: We corrected this.

Detailed answers to specific comments from Reviewer 3:

1. Reviewer 3 states: “This is a robust study of the genetic contribution of SRS2 to BIR. The data are well presented and the authors’ arguments are easy to follow. Genetically, the experiments seem pretty solid....”

Answer: We are very thankful to the reviewer for this good concrete assessment of our study. We appreciate that the reviewer understands the power of yeast genetics and we believe that (as detailed below) a combination of genetics and molecular biology are sufficient to address all of the goals of this paper.

2. Reviewer 3 states: “Where I have more difficulty is the inference of mechanism solely through genetic dissection. Yes, the data might be consistent with the model presented but no attempt is made to use cell biological or biochemical methods to probe the model beyond using standard 2D gels. In similar studies on repair/recombination pathways in human cells the authors would be expected to use an array of cellular and biochemical techniques to identify the localization and kinetics of proteins involved in such a process. As a result, the approach and findings of the work here come across as esoteric and based on a considerable amount of inference.”

Answer: Since the main focus of our research is DNA repair, the main output of our studies includes the description of DNA repair outcomes that can be assayed genetically or physically by direct analysis of DNA. Baker’s yeast is the best system to study eukaryotic DNA repair, and especially repair of DSBs using direct methods. It is easy in yeast to initiate DSBs synchronously, to visualize breaks by DNA analysis, to detect DSB end resection, and to follow the kinetics of DSB repair by using native, denaturing or CHEF gel electrophoresis, and Southern blot analysis. In addition, 2D gel electrophoresis and electron microscopy can be used to analyze the structure of DNA repair intermediates. However, the direct methods of DNA analysis available in yeast are not always available in other experimental systems, including mammalian cells, where indirect methods of analysis must often be employed. Genome size and lack of DSB induction synchrony makes physical analysis of recombination intermediates impossible. Thus other methods are often used to understand the role of DNA repair enzymes. For example, the appearance of γ H2AX signals is often used as surrogate for DSBs; the RPA foci are used to follow DSB end resection, while loading of Rad51 or Rad54 allows visualization of recombination progression. Not surprisingly, the usage of indirect approaches often require that several methods are employed to confirm every conclusion. In the recent years, several reporter DSB repair assays^{37,38}, analogous to those used in yeast, have been developed in mammalian cells. Development of these new reporter systems allowed usage of direct methods in mammalian cells as well.

Our research presented here is based on the results of many studies that were performed in other labs, including investigation of Srs2, Rad51 and other proteins by using biochemical, cell biology, genetic and other methods. For example a host of biochemical studies^{1,2,4,34,35} led to identification of several structural domains of Srs2 and to characterize their roles in disassembly of Rad51 filament and in DNA unwinding. Also, it was reported that DSB induction in *srs2 Δ* mutants leads to accumulation of Rad51²⁷ and RPA foci²⁷ along large chromosomal regions, which was proposed to promote cell death by interfering with DNA repair synthesis or by persistent checkpoint signaling. However, these observations (made by using various methods) could not explain our observations, which include: (i) the trapping of donor and recipient chromosomes together and the compromised survival in cells containing the intact copy of a broken

chromosome; (ii) the dependence of cell survival following DSB induction on the helicase rather than on Rad51 strippase activity, and (iii) the role of Mus81 and Yen1 in the resolution of toxic molecules and promoting cell survival. Here, using a combination of direct physical approaches, including two-dimensional (2D) gel electrophoresis and electron microscopy (EM), we were able for the first time, to detect and to determine the structure of toxic joint molecules that are accumulated in the absence of Srs2, and which have been previously postulated but never directly visualized. To our knowledge this structure that we call “rubble” was not previously observed and it is different from all other structures previously followed by 2D gel electrophoresis, including Holliday Junction, migrating D-loop, replication forks, and firing replication origins, and it is specific for *srs2Δ* mutant. The detection and analysis of these structures was instrumental in uncovering the anti-toxic role of Srs2.

Finally, the methods we employed and the experimental system we used allowed us not only to produce the data consistent with our model, but also to probe this model further by testing its main predictions. In particular, we predicted that the need for Srs2 in BIR could be bypassed in *rad55Δ* or *rfa1-t33* mutants where the Rad51 filament is intrinsically unstable. This prediction was based on many biochemical, and genetic studies conducted by other labs, which allowed us simply to use these mutations as a powerful tool for testing our model. Using these tools we were able not only to test our model, but also to make new findings. For example, we observed that the Rad51 filament assembled in the absence of Rad55 and Srs2 differs from the one assembled in the situation when these proteins are present, and therefore, both Srs2 and Rad55 are required for the assembly of an optimum filament.

Overall, we believe that the experimental system we used as well as the methods that we employed were optimal for achieving the goals of this study, and while some new cell biology and biochemistry experiments could be used in the future, their goals, seem to be outside of the scope of this paper.

Reference:

1. Colavito, S. et al. Functional significance of the Rad51-Srs2 complex in Rad51 presynaptic filament disruption. *Nucleic Acids Res* **37**, 6754-64 (2009).
2. Antony, E. et al. Srs2 disassembles Rad51 filaments by a protein-protein interaction triggering ATP turnover and dissociation of Rad51 from DNA. *Mol Cell* **35**, 105-15 (2009).
3. Nguyen, J.H.G. et al. Differential requirement of Srs2 helicase and Rad51 displacement activities in replication of hairpin-forming CAG/CTG repeats. *Nucleic Acids Res* **45**, 4519-4531 (2017).
4. Liu, J. et al. Srs2 promotes synthesis-dependent strand annealing by disrupting DNA polymerase delta extending D-loops. *eLife* **10.7554/eLife 22195**(2017).
5. Wright, W.D. & Heyer, W.D. Rad54 functions as a heteroduplex DNA pump modulated by its DNA substrates and Rad51 during D loop formation. *Mol Cell* **53**, 420-32 (2014).

6. Deem, A. et al. Break-induced replication is highly inaccurate. *PLoS Biol* **9**, e1000594 (2011).
7. Thangavel, S. et al. DNA2 drives processing and restart of reversed replication forks in human cells. *J Cell Biol* **208**, 545-62 (2015).
8. Ahuja, A.K. et al. A short G1 phase imposes constitutive replication stress and fork remodelling in mouse embryonic stem cells. *Nat Commun* **7**, 10660 (2016).
9. Zellweger, R. et al. Rad51-mediated replication fork reversal is a global response to genotoxic treatments in human cells. *J Cell Biol* **208**, 563-79 (2015).
10. Neelsen, K.J., Chaudhuri, A.R., Follonier, C., Herrador, R. & Lopes, M. Visualization and interpretation of eukaryotic DNA replication intermediates in vivo by electron microscopy. *Methods Mol Biol* **1094**, 177-208 (2014).
11. Aboussekhra, A. et al. RADH, a gene of *Saccharomyces cerevisiae* encoding a putative DNA helicase involved in DNA repair. Characteristics of radH mutants and sequence of the gene. *Nucleic Acids Res* **17**, 7211-9 (1989).
12. Rong, L., Palladino, F., Aguilera, A. & Klein, H.L. The hyper-gene conversion hpr5-1 mutation of *Saccharomyces cerevisiae* is an allele of the SRS2/RADH gene. *Genetics* **127**, 75-85 (1991).
13. Aboussekhra, A., Chanet, R., Adjiri, A. & Fabre, F. Semidominant suppressors of Srs2 helicase mutations of *Saccharomyces cerevisiae* map in the RAD51 gene, whose sequence predicts a protein with similarities to procaryotic RecA proteins. *Mol Cell Biol* **12**, 3224-34 (1992).
14. Barbour, L. & Xiao, W. Regulation of alternative replication bypass pathways at stalled replication forks and its effects on genome stability: a yeast model. *Mutat Res* **532**, 137-55 (2003).
15. Watts, F.Z. The role of SUMO in chromosome segregation. *Chromosoma* **116**, 15-20 (2007).
16. Lambert, S. et al. Homologous recombination restarts blocked replication forks at the expense of genome rearrangements by template exchange. *Mol Cell* **39**, 346-59 (2010).
17. Robert, T., Dervins, D., Fabre, F. & Gangloff, S. Mrc1 and Srs2 are major actors in the regulation of spontaneous crossover. *EMBO J* **25**, 2837-46 (2006).
18. Le Breton, C. et al. Srs2 removes deadly recombination intermediates independently of its interaction with SUMO-modified PCNA. *Nucleic Acids Res* **36**, 4964-74 (2008).
19. Burgess, R.C. et al. Localization of recombination proteins and Srs2 reveals anti-recombinase function in vivo. *J Cell Biol* **185**, 969-81 (2009).
20. Kerrest, A. et al. SRS2 and SGS1 prevent chromosomal breaks and stabilize triplet repeats by restraining recombination. *Nat Struct Mol Biol* **16**, 159-67 (2009).
21. Urulangodi, M. et al. Local regulation of the Srs2 helicase by the SUMO-like domain protein Esc2 promotes recombination at sites of stalled replication. *Genes Dev* **29**, 2067-80 (2015).
22. Gangloff, S., Soustelle, C. & Fabre, F. Homologous recombination is responsible for cell death in the absence of the Sgs1 and Srs2 helicases. *Nat Genet* **25**, 192-4 (2000).

23. Klein, H.L. Mutations in recombinational repair and in checkpoint control genes suppress the lethal combination of srs2Delta with other DNA repair genes in *Saccharomyces cerevisiae*. *Genetics* **157**, 557-65 (2001).
24. Aguilera, A. & Klein, H.L. Genetic control of intrachromosomal recombination in *Saccharomyces cerevisiae*. I. Isolation and genetic characterization of hyper-recombination mutations. *Genetics* **119**, 779-90 (1988).
25. Liu, J. et al. Rad51 paralogues Rad55-Rad57 balance the antirecombinase Srs2 in Rad51 filament formation. *Nature* **479**, 245-8 (2011).
26. Heude, M., Chanet, R. & Fabre, F. Regulation of the *Saccharomyces cerevisiae* Srs2 helicase during the mitotic cell cycle, meiosis and after irradiation. *Mol Gen Genet* **248**, 59-68 (1995).
27. Yeung, M. & Durocher, D. Srs2 enables checkpoint recovery by promoting disassembly of DNA damage foci from chromatin. *DNA Repair (Amst)* **10**, 1213-22 (2011).
28. Vasianovich, Y. et al. Unloading of homologous recombination factors is required for restoring double-stranded DNA at damage repair loci. *EMBO J* **36**, 213-231 (2017).
29. Jain, S. et al. A recombination execution checkpoint regulates the choice of homologous recombination pathway during DNA double-strand break repair. *Genes Dev* **23**, 291-303 (2009).
30. Vaze, M.B. et al. Recovery from checkpoint-mediated arrest after repair of a double-strand break requires Srs2 helicase. *Mol Cell* **10**, 373-85 (2002).
31. Jain, S., Sugawara, N. & Haber, J.E. Role of Double-Strand Break End-Tethering during Gene Conversion in *Saccharomyces cerevisiae*. *PLoS Genet* **12**, e1005976 (2016).
32. Chung, W.H., Zhu, Z., Papusha, A., Malkova, A. & Ira, G. Defective resection at DNA double-strand breaks leads to de novo telomere formation and enhances gene targeting. *PLoS Genet* **6**, e1000948 (2010).
33. Morel, P., Hejna, J.A., Ehrlich, S.D. & Cassuto, E. Antipairing and strand transferase activities of *E. coli* helicase II (UvrD). *Nucleic Acids Res* **21**, 3205-9 (1993).
34. Krejci, L. et al. DNA helicase Srs2 disrupts the Rad51 presynaptic filament. *Nature* **423**, 305-9 (2003).
35. Veaute, X. et al. The Srs2 helicase prevents recombination by disrupting Rad51 nucleoprotein filaments. *Nature* **423**, 309-12 (2003).
36. Keyamura, K., Arai, K. & Hishida, T. Srs2 and Mus81-Mms4 Prevent Accumulation of Toxic Inter-Homolog Recombination Intermediates. *PLoS Genet* **12**, e1006136 (2016).
37. Chandramouly, G. et al. BRCA1 and CtIP suppress long-tract gene conversion between sister chromatids. *Nat Commun* **4**, 2404 (2013).
38. Jasin, M. & Haber, J.E. The democratization of gene editing: Insights from site-specific cleavage and double-strand break repair. *DNA Repair (Amst)* **44**, 6-16 (2016).

REVIEWERS' COMMENTS:

Reviewer #1 (Remarks to the Author):

The authors have addressed my points fully. This is a very nice study that provides considerable insights into the regulatory roles of Srs2 in homologous recombination pathways. I am enthusiastic about its acceptance.

Reviewer #2 (Remarks to the Author):

I am satisfied with the authors' response to my concerns and think that the manuscript is much improved.